# Development of pedotransfer functions for water retention in tropical mountain soilscapes: Spotlight on parameter tuning in machine learning

Anika Gebauer[1], Monja Ellinger[1], Victor M. Brito Gomez[2], Mareike Ließ[1]

[1]Department Soil System Science, Helmholtz Centre for Environmental Research – UFZ, Halle (Saale), Germany
[2]Departamento de Recursos Hídricos y Ciencias Ambientales, Facultad de Ciencias Agropecuarias, Universidad de Cuenca, Cuenca, Ecuador

*Correspondence to*: Anika Gebauer (anika.gebauer@ufz.de)

**Abstract.** Machine learning algorithms are good in computing non-linear problems and fitting complex composite functions,
which makes them an adequate tool to address multiple environmental research questions. One important application is the development of pedotransfer functions (PTF). This study aims to develop water retention PTFs for two remote tropical mountain regions of rather different soil-landscapes, dominated by (1) peat soils and soils under volcanic influence with high organic matter contents, and (2) tropical mineral soils. Two tuning procedures were compared to fit boosted regression tree models: (1) tuning by grid search, which is the standard approach in pedometrics, and (2) tuning by differential evolution
optimization. A nested cross-validation approach was applied to generate robust models. The developed area-specific PTFs outrival other more general PTFs. Furthermore, the first PTF for typical soils of Páramo landscapes (Ecuador), i.e. organic soils under volcanic influence, is presented. Overall, results confirmed the differential evolution algorithm's high potential for tuning machine learning models. While models based on tuning by grid search roughly predicted the response variables' mean for both areas, models applying the differential evolution algorithm for parameter tuning explained up to 25 times more of the
response variables' variance.

## 1 Introduction

Machine learning algorithms are good at fitting highly complex non-linear functions (Witten et al., 2011). Major application fields in soil science investigate the soils' spatial variability (Heung et al., 2016), relate data from soil sensing to soil properties (Viscarra Rossel et al., 2016), or develop pedotransfer functions (PTFs, Botula et al., 2014; Van Looy et al., 2017). McBratney
et al. (2019) give a timeline on developments in pedometrics, that refer to machine learning in multiple applications.

Pedotransfer functions derive laborious and complex soil parameters (response variables) from more readily available soil properties (predictor variables). Most PTFs are developed to predict soil hydraulic properties. Reviews on the involved methodologies are provided by Pachepsky and Rawls (2004), Shein and Arkhangel'skaya (2006) and Vereecken et al. (2010). Machine learning algorithms applied for PTF development include e.g. support vector machines (Lamorski et al., 2008),

artificial neural networks (Haghverdi et al., 2012) and regression trees (Tóth et al., 2015). According to Van Looy et al. (2017), most PTFs are developed for mineral soils, while PTFs applicable to organic soils or soils with specific properties like volcanic ash soils are highly underrepresented. Patil and Singh (2016) and Botula et al. (2014) provide reviews of hydrological PTFs for mineral soils of certain tropical and temperate regions. With particle size distribution (PSD) being the basic input parameter to derive soil hydrologic properties, most PTFs also use bulk density (BD) and soil organic carbon content (SOC) as predictors.

As summarized by Patil and Singh (2016), the application of existing hydrological PTFs is often restricted because of two reasons: (1) The majority of PTFs are developed on soils that developed under certain conditions. Often these PTFs cannot be applied in other regions as the site-specific soil-forming conditions can cause considerable differences in physical and chemical soil properties. This is for example demonstrated by Botula et al. (2012) and Moreira et al. (2004) who were able to show that, when applied to independent tropical soil data, existing temperate PTFs perform worse than existing tropical PTFs. (2) The

applicability of existing PTFs is further restricted by the required input data. As stated by Morris et al. (2019) hydraulic PTFs developed on mineral soils are often inapplicable to organic soils. The measurement of the predictor variable PSD may be hampered by high organic matter contents, and organic soils may not include sufficient mineral soil material to justify PSD analysis at all. Overall, there is only a small number of PTFs that were developed for organic soils and most of them are based on data from specific temperate regions and rely on very specific predictor variables. Korus et al. (2007) for example related

the water retention of polish peat soils to the predictors ash content, specific surface area, BD, pH, and iron content. In Finish peat soils semi-empirical water retention PTFs were developed on different predictors including BD, sampling depth and botanical residues (Weiss et al., 1998). Even though it was never intended to be used for predictions, Rocha Campos et al. (2011) provide the only regression model known to us, which relates the soil hydrologic parameters of tropical organic soils to independent variables (fiber content, mineral material, BD and organic matter fractions).

The application of machine learning algorithms requires them to be adjusted to the specific modeling problem by parameter tuning. Tuning parameter values cannot be calculated analytically, so in soil science applications grid search is often used as a standard technique (e.g. Babangida et al., 2016; Khlosi et al., 2016; Twarakavi et al., 2009). It works by testing a number of predefined parameter values or combinations of parameter values to finally choose the best. Accordingly, the predominant part of the multivariate parameter space cannot be searched in the case of continuous parameters and the optimum might not be

found. To overcome this limitation, mathematical optimization is a promising alternative. Commonly applied optimization algorithms include artificial bee colony, simulated annealing, particle swarm optimization, the nelder-mead method, bayesian optimization or evolutionary and genetic strategies. Their applications range from pattern recognition (e.g. Jayanth et al., 2015; Liu and Huang, 1998), through solving combinatorial problems (e.g. Kang-Ping Wang et al., 2003; Reeves, 1993) to parameter tuning in machine learning (e.g. Imbault and Lebart, 2004; Ozaki et al., 2017). We would like to particularly emphasize the

differential evolution algorithm. Price et al. (2005), who compared it to various other optimization algorithms, were able to show that it usually leads to better results and comparatively low computing times. This has been confirmed by the results of Chen et al. (2017) who compared differential evolution to particle swarm optimization and a genetic algorithm in landslide modeling, and Yin et al. (2018) who compared differential evolution to simulated annealing, particle swarm optimization,

artificial bee colony and genetic algorithms in geotechnical engineering. It is also able to outperform bayesian approaches in certain applications. Comparisons of both algorithms led to contradictory results: while some studies found bayesian approaches to be superior (e.g. Carr et al., 2016) others reported the opposite result (e.g. Schmidt et al., 2019). The differential evolution algorithm was applied to diverse optimization problems including the prediction of stable metallic clusters (Yang et al., 2018), the navigation of robots (Martinez-Soltero and Hernandez-Barragan, 2018), the classification of microRNA targets (Bhadra et al., 2012), parity-P problems (Slowik and Bialko, 2008) or the parameter tuning of machine learning models trained to e.g. predict landslides (Tien Bui et al., 2017). In soil-related research questions it was applied e.g. to optimize parameters of geostatistical models (Brus et al., 2016; Wadoux et al., 2018) or to optimize parameters defining the shape of well-known soil water retention curves (Maggi, 2017; Ou, 2015). However, in pedometrics, applications for parameter tuning in machine learning are scarce (e.g. Gebauer et al., 2019).

This study first of all aims to develop water retention PTFs for two tropical soil-landscapes dominated by (1) peat soils and soils under volcanic influence with high organic matter contents, like they are commonly occurring in Páramo regions (Ecuador), and (2) tropical soils of a dry climate. Currently, PTFs suitable for the soils of these regions are at best very few, if any, at all. The parameter tuning technique is assumed to affect the performance of the machine learning based PTFs. This is why our second and equally important aim is to compare the differential evolution algorithm to grid search. On average, different machine learning algorithms perform equally well (Wolpert, 2001). We have chosen to fit boosted regression tree models, because we assume that the pre-eminence of optimization for parameter tuning in machine learning will particularly show when applying it to a machine learning algorithm that requires not only the fitting of discrete-valued parameters, but also the fitting of numerous continuous parameters.

## 2 Material and methods

### 2.1 Research areas

The two investigated soil-landscapes are situated in southern Ecuador (Fig. 1). The Quinuas catchment encompasses an area of about 93 km$^2$, including parts of the Cajas National Park (Fig. 1c). Being located between 3000 and 4400 m above sea level (a.s.l.), the mean annual temperature ranges between 5.3 and 8.7 °C with no seasonality (Carrillo-Rojas et al., 2016). With peaks in March/May and October (Celleri et al., 2007) mean annual precipitation varies between 900 and 1600 mm (Crespo et al., 2011). Due to volcanic ash deposits and the cold and wet climate, soils with a low bulk density and high SOC contents are typical (Buytaert et al., 2007). The Quinuas catchment can be allocated to the Páramo ecosystem (Guio Blanco et al., 2018), which plays a major role in the water supply of the inter-Andean region (Buytaert et al., 2006a, 2006b, 2007).

The Laipuna dry forest region is part of the "Laipuna Conservation and Development Area" and covers approximately 16 km² (Fig. 1d). Its temperature profile shows little seasonal variability, while there is a rain period from January to May. Depending on the altitude ranging between 400 and 1500 m a.s.l., the mean annual temperature varies between 16 and 23 °C and mean annual precipitation between 540 and 630 mm (Peters and Richter, 2011b, 2011a). Additionally, the El Niño-Southern

Oscillation influences the area (Bendix et al., 2003, 2011). Laipuna is part of an ecosystem with high biodiversity and many endemic species (Best and Kessler, 1995; Linares-Palomino et al., 2009), which are strongly adapted to the ecosystem and may be threatened by possible climate-induced changes of the water supply.

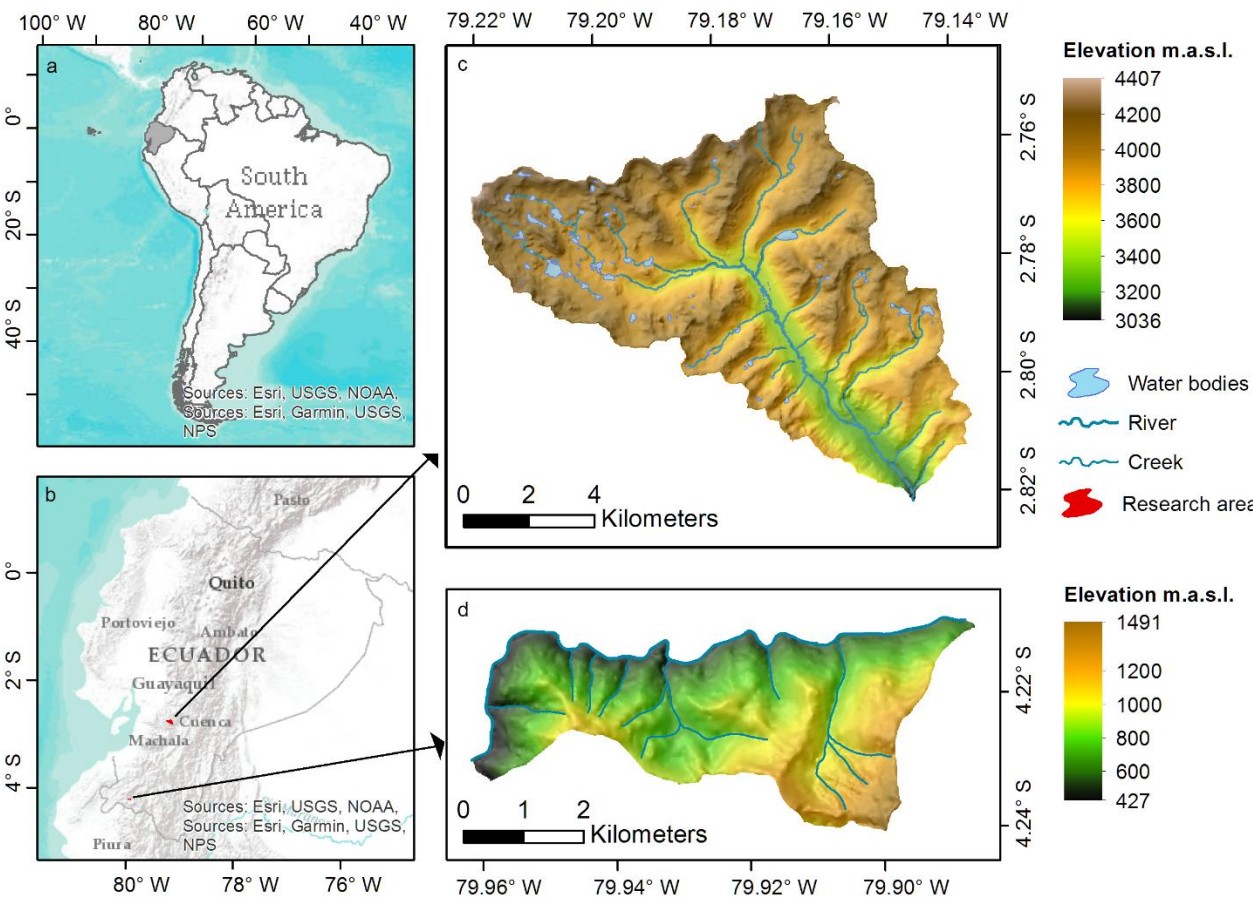

**Fig. 1. Research areas.** a) Ecuador within South America, b) Research areas within Ecuador, c) Quinuas, d) Laipuna (overlaid hillshading with a light source from north). Adapted from Ließ (2015). Topographical data was used with permission from the Ecuadorian Geographical Institute (2013, national base, scale 1:50.000), further GIS data was provided by NCI and ETAPA.

## 2.2 Soil data

To ensure representative datasets for both areas, sampling sites were selected using the algorithm "QC-arLUS" (Ließ, 2015). The algorithm divides a research area into strata, which represent characteristic landscape structures. Actual sampling site selection per stratum is limited to the accessible area. For Quinuas and Laipuna two sampling sites were chosen per landscape stratum, resulting in 46 sites for Quinuas and 55 for Laipuna. Soil profiles were excavated at these sites. However, due to laboratory constraints, samples for the determination of soil water retention were taken from the topsoil, only. Water retention at pF 0 ($-10^0$ hPa), 0.5 ($-10^{0.5}$ hPa), 1.5 ($-10^{1.5}$ hPa) and 2.5 ($-10^{2.5}$ hPa) was measured in three replicate samples according to

DIN EN ISO 11274:2014-07: hanging water columns of increasing length were applied to undisturbed steel core samples of 100 cm³. The high amount of organic matter in the Quinuas soil samples prevented water retention measurements at higher pF values. BD and SOC content were used as predictors for both research areas to develop the water retention PTFs, PSD only for Laipuna. For BD measurements according to DIN EN ISO 11272:2017-07, undisturbed samples (three replicates) were oven-dried at 105 °C for three days. Disturbed samples (three replicates), were tested for carbonates with 10% hydrochloric

acid, sieved to 2 mm, and ground before SOC content determination using dry combustion (DIN EN 15936:2012-11). Disturbed samples from Laipuna were oven-dried at 40 °C, sieved to 2 mm and PSD was determined according to DIN ISO 11277:2002-08 in two (sand fractions) and three (clay and silt fractions) replicate samples. Measurements distinguish the following particle size classes: clay (< 2 µm), fine silt (2 – 6.3 µm), medium silt (6.3 – 20 µm), coarse silt (20 – 62 µm), fine sand (62 – 200 µm), medium sand (200 – 630 µm) and coarse sand (630 – 2000 µm). The high soil organic matter contents

prevented PSD measurements in Quinuas. As suggested by Guio Blanco et al. (2018), models built on the Quinuas dataset could be improved by treating samples from mineral soils as outliers and removing them. For both research areas, it was decided to remove only data pairs of response and predictor variables that were identified as multivariate outliers. Tests for multivariate outliers were done by building hierarchical clusters using the "hclust" function from R-package "fastcluster" (Müller, 2018), version 3.4.4. To enhance comparability, models were trained on response variables scaled to the range [0, 1]

following Eq (1):

$$x_j[0, 1] = \frac{x_j - \min(x)}{\max(x) - \min(x)}, \tag{1}$$

where x is the vector of the response variables of length *j*.

### 2.3 Boosted regression trees

    The BRT algorithm combines the machine learning techniques regression trees and boosting. Tree models use decision rules,

which involve the predictor variables, to recursively partition the response variable data into increasingly similar subgroups until terminal nodes are reached (Kuhn and Johnson, 2013). For each subgroup, the response variable values of the terminal regression tree nodes are averaged to be used for the prediction (James et al., 2017). The boosting machine learning technique improves the overall model accuracy by combining a number of simple models (Witten and Frank, 2005). To develop the PTFs, BRT models were trained using the "gbm" R-package, version 2.1.3 (Ridgeway, 2017), which is based on Friedman's

(2002) stochastic gradient boosting. This boosting technique iteratively fits a number of simple regression tree models to random training data subsets. In each iteration a new regression tree is added to the model until many simple regression trees form a linear combination: the final BRT model. Each tree that is added, improves the overall model performance. The first tree improves model performance the most. The further regression trees are fitted with emphasis on observations that are predicted poorly by the existing model. To apply a BRT model, usually up to four parameters are tuned: number of trees

(n.trees), shrinkage, interaction depth, and bag fraction (e.g. Ottoy et al., 2017; Wang et al., 2017; Yang et al., 2016). Elith et al. (2008) provide a detailed analysis of their function: The n.trees parameter describes the number of regression tree models

to be iteratively fitted. Shrinkage defines the model's learning rate by scaling the outcome of each simple regression tree, thereby controlling its contribution to the final model. The interaction depth parameter controls the number of splits in each tree to divide the response variable data into subgroups. The bag fraction parameter determines the size of the randomly selected data subsets. This is able to reduce the risk of overfitting (Friedman, 2002), but may lower the model robustness (Elith et al., 2008). To develop PTFs for Quinuas and Laipuna, these four parameters were tuned following the steps described in Section 2.4 and 2.5.

## 2.4 Parameter tuning

Parameter tuning was done in two different ways: (1) by grid search and (2) by optimization applying the differential evolution algorithm. Grid search compares a certain number of predefined $k$ dimensional parameter vectors. In order to reduce computing time, the number of predefined values of the $k = 4$ parameters was limited to five for each. The selected values were based on the recommendations of Elith et al. (2008) and Ridgeway (2012). They are summarized in Table 1. Finally, $5^k$ different combinations of tuning parameter values, i.e. 625, were compared.

In contrast to this, the differential evolution optimization algorithm, developed by Storn and Price (1995), is able to search the multivariate space between defined upper and lower parameter limits. The parameter values are optimized by minimizing an objective function, which defines their suitability. The objective function is allowed to be stochastic and noisy and does not need to be differentiable or continuous (Mullen et al., 2011). Following the evolutionary theory, this is done by repeating three steps for $i$ iterations: mutation, crossover, and selection (Fig. 2). At first, an initial parent population of a number ($v$) of $k$-dimensional parameter vectors is generated randomly. With each iteration $i$, these vectors are changed by mutation and randomly mixed by crossover to generate a new population. Selection compares the objective function values belonging to the parent and the new vector to decide whether a new vector replaces its parent vector. Differential evolution was applied using the R-package "DEoptim", version 2.2.4 (Ardia et al., 2016). For each tuning parameter, optimization limits correspond to the maximum and minimum grid search values (Table 1). The number of vectors of size $k = 4$ tuning parameters was set to $v = 100$. The R-package's default mutation strategy was used, which changes each parent vector by adding two summands: (1) the difference between two random parent vectors and (2) the difference between the vector to be perturbed and the best vector found in the parent population. Summands were scaled by the factor 0.8. For crossover, the probability of randomly mixing the parent and the mutated vectors' elements was set to 50%. To reduce computing time, the optimization process was stopped either after $i_{max} = 10$ iterations without improving the objective function or a maximum number of 200 iterations. two- the selection step, the discrete tuning parameter values (n.trees, interaction depth) were rounded, as the differential evolution algorithm treats all values as real numbers during mutation and crossover. To select the final tuning parameter values, grid search and differential evolution both minimized the cross validated $RMSE_T$ as objective function. $RMSE_T$ calculation is explained in Section 2.5.

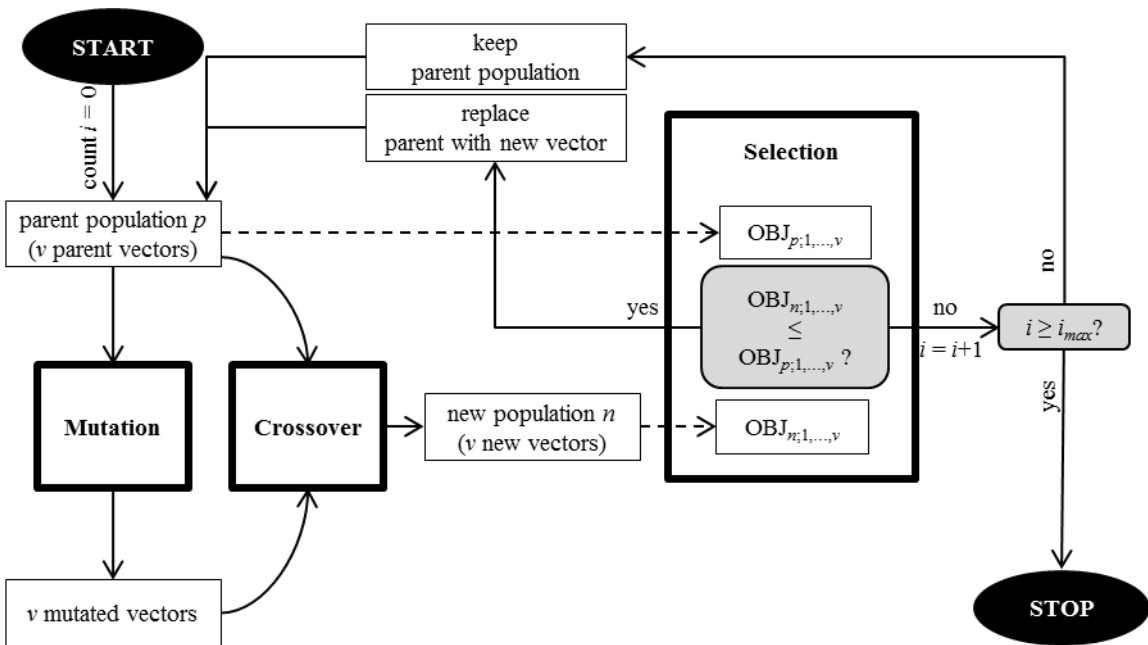

Fig. 2. Flowchart of the differential evolution algorithm. OBJ = objective function, $p$ = parent population, $n$ = new population, $i$ = iteration, $i_{max}$ = maximum number of iterations, $v$ = number of vectors. Adapted from Gebauer et al. (2019).

Table 1. Tuning parameter values to be tested by grid search and optimization limits required by the differential evolution algorithm.

| Tuning parameter | Grid search values | Differential evolution limits | |
|---|---|---|---|
| n.trees | 100; 1000; 2000; 3000; 4000 | 100 | 4000 |
| shrinkage | 0.001; 0.005; 0.01; 0.05; 0.1 | 0.001 | 0.1 |
| interaction depth | 1; 2; 3; 4; 5 | 1 | 5 |
| bag fraction | 0.5; 0.6; 0.7; 0.8; 0.9 | 0.5 | 0.9 |

## 2.5 Performance evaluation

In order to build robust models, we followed a nested cross-validation (CV) approach. Stratified five-fold CV was applied for two purposes: (1) to conduct robust parameter tuning on resampled data subsets by either grid search or the differential evolution algorithm, and (2) to evaluate the final performance of models built on tuned parameter values. CV provides error metrics with good bias and variance properties, is beneficial for small datasets and avoids overfitting (Arlot and Celisse, 2009; James et al., 2017). Following the steps shown in Fig. 3, the stratified five-fold CV was implemented with five repetitions for model evaluation and one repetition for parameter tuning. In Step 1, the complete dataset (n = 100%) was split into five folds with each of them (n = 20%) once used as the test set, leaving the remaining folds as the model training set. For resampling in parameter tuning (Step 2), each model training set was again subdivided, similar to Step 1. Each tuning parameter vector in grid search and the differential evolution algorithm was evaluated by the cross-validated $RMSE_T$ (Step 3, Step 4). By

comparing the $RMSE_T$, the best vector of tuning parameter values for each model evaluation training set was selected and applied (Step 5, Step 6). To assess model performance, the coefficient of determination ($R^2_E$) and the root mean squared error

($RMSE_E$) of model evaluation were calculated by predicting the associated test set data (Step 7). To divide the datasets into folds, the function "partition_cv_strat" from R-package "sperrorest", version 2.0.0 (Brenning et al., 2017) was applied, with three equal probability strata of the response variable's density function.

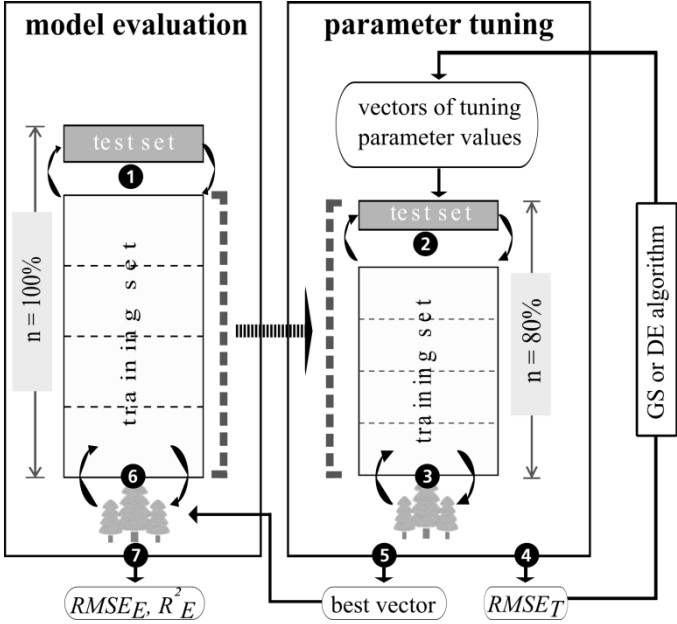

**Fig. 3. Nested cross-validation approach comprising model evaluation and parameter tuning.** Adapted from Guio Blanco et al. (2018).
The tree icons symbolize BRT models, which are repeatedly (circular arrows) trained and tested on different data sets. The numbers within black circles belong to the steps described in Section 2.5. $RMSE_T$ = root mean squared error of parameter tuning, $RMSE_E$ = root mean squared error of model evaluation, $R^2_E$ = coefficient of determination of model evaluation, GS = grid search, DE = differential evolution.

## 2.6 Comparison to existing PTFs

To further assess the developed BRT PTFs, it was decided to compare their results to predictions resulting from the application
of existing PTFs. PTFs that were developed on different datasets, but under conditions as similar as possible to those of Quinuas and Laipuna, were selected from the literature. If more than one PTF was provided per study, the one with the best reported performance was applied. For Laipuna, seven PTFs (Table 2) were chosen based on four criteria: (1) developed for tropical soils, (2) similar predictor variables, (3) regression equation provided and (4) included in the peer-reviewed Clarivate Analytics' Web of Science database. To be able to apply the readily available equations with predictors of the Laipuna dataset,
it was necessary to convert the determined soil texture classes to the respective USDA classes. Following the approach of Shang (2013), texture conversion was done using spline interpolation. Because of different predictor variables, it is difficult to find organic PTFs applicable to the Quinuas dataset. An exhaustive literature search revealed only the PTF of Boelter (1969), who related water retention at pF 0 to BD for temperate peat soils in northern Minnesota.

 **3 Results and discussion**

**3.1 Model input**

For Laipuna, data pairs of four sampling sites were identified as multivariate outliers. After removing them, the datasets contained predictor and response variables of 51 and 46 sampling sites for Laipuna and Quinuas, respectively. A summary of the remaining unscaled data is shown in Fig. 4 and 5. As expected both areas show huge differences regarding the values of
response and predictor variables. BD values in Quinuas range from 64 to 807 kg m$^{-3}$, while SOC values vary between 8.8 and 46.4 wt.%. SOC values are normally distributed, while BD data display a positive skew. Averagely decreasing by 22%, water retention ranges from 0.25 (pF 2.5) to 0.94 cm$^3$ cm$^{-3}$ (pF 0). While the data displays a positive skew for pF 0, the data distribution for the other pF values shows a negative skew. For Laipuna, BD ranges between 1157 and 1727 kg m$^{-3}$, displaying a distribution with a positive skew. The SOC content is normally distributed and varies between 0.4 and 3.8 wt. %. Clay content
ranges between 17 and 48 %, silt between 24 and 45 % and sand between 14 and 50 %. Especially fine and medium silt show skewed distributions. Water retention values are ranging between 0.25 (pF 2.5) and 0.61 cm$^3$ cm$^{-3}$ (pF 0). On average, they decrease by 37 % with increasing water tension. Data is skewed positively for pF 0 and negatively for pF 0.5.

Quinuas soils go along with the low density, porous soils, rich in organic material, that are found throughout the Paute river basin (Buytaert et al., 2007; Poulenard et al., 2003). Loosely bedded volcanic ash deposits explain the low bulk density values
(Buytaert et al., 2007). High SOC contents are caused by low redox potentials and the presence of organometallic complexes inhibiting degradation processes (Buytaert et al., 2006a). Comparatively high water retention values can be attributed to the porous structure of Páramo soils being able to retain a lot of water (Buytaert et al., 2007). High contents of soil organic matter are associated with soils characterized by a high water holding capacity (Buytaert et al., 2007), which explains the relatively small decrease in water retention with increasing water tension. Measured BD and SOC contents are in accordance with data
observed for other Páramo regions (e.g. Buytaert et al., 2007, 2006b). The water retention values are comparable to data obtained in other Páramo areas (Buytaert et al., 2005) and soils with high organic matter contents (Schwärzel et al., 2002, 2006). Extreme values in BD and water retention (Fig. 4 and Fig. 5) correspond to less frequent mineral soils with much lower SOC contents (Guio Blanco et al., 2018). Expecting these values to be reliable, they were not removed from the model input. The BD and SOC values measured in Laipuna correspond to other dry forest ecosystems (e.g. Conti et al., 2014; de Araújo
Filho et al., 2017; Singh et al., 2015), whereas the PSD shows higher clay contents compared to the dry forest soils investigated by Cotler and Ortega-Larrocea (2006), Jha et al. (1996) and Sagar et al. (2003). Measured water retention values are higher than those obtained in a tropical dry forest in Brazil (Vasques et al., 2016), probably caused by the higher clay content enhancing the water holding capacity.

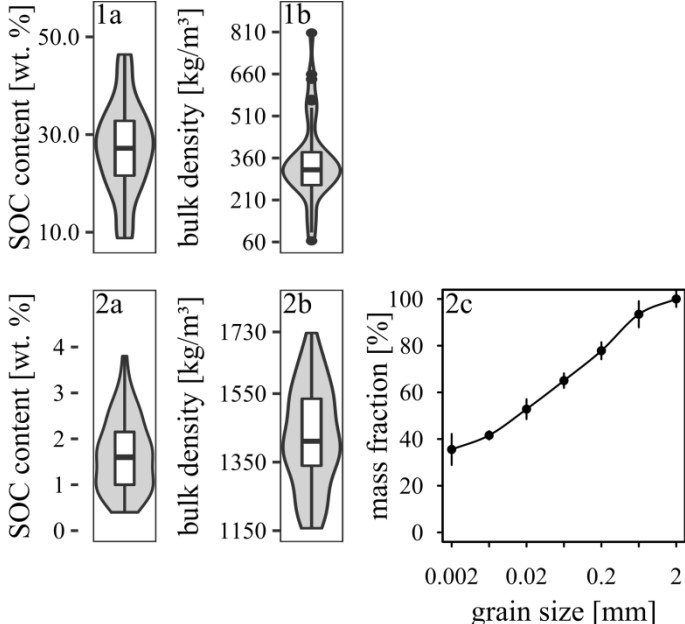

**Fig. 4. Predictor variables.** 4.1 Quinuas (46 samples), 4.2 Laipuna (51 samples), a) SOC content, b) bulk density, c) particle size distribution displayed as a cumulative distribution function (mean values with standard deviation). High organic matter contents prevented measurements of the particle size distribution in Quinuas.

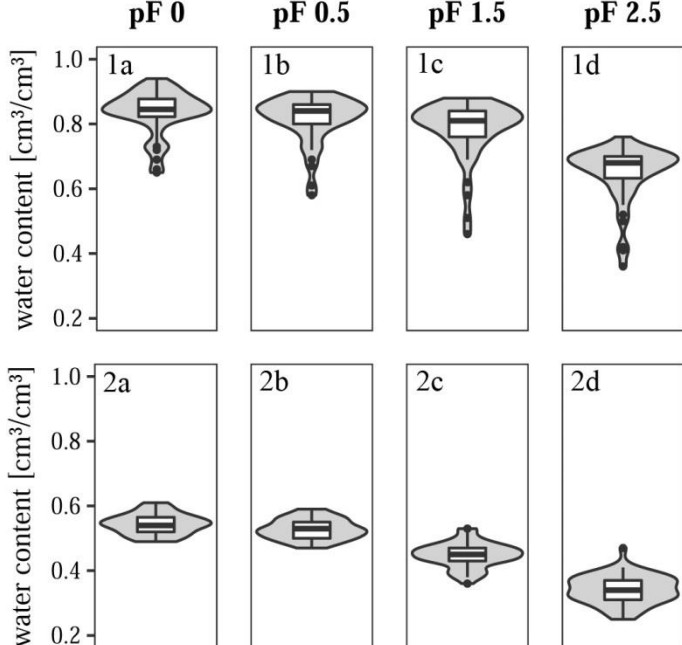

**Fig. 5. Response variables.** 5.1 Quinuas (46 samples), 5.2 Laipuna (51 samples), water retention at a) pF 0, b) pF 0.5, c) pF 1.5, d) pF 2.5.

## 3.2 Model performance

The performance of the final models built on parameters selected by grid search and the differential evolution algorithm is demonstrated by the error metrics $R^2_E$ and $RMSE_E$ in Fig. 6 (Quinuas) and 7 (Laipuna) as well as by scatterplots comparing observed and predicted water retention values in Fig. 8 (Quinuas) and 9 (Laipuna). Error metrics as well as scatterplots are based on response variables scaled to the range [0, 1]. All grid search models resulted in very similar mean $RMSE_E$ values between 0.20 (pF 1.5) and 0.22 (pF 0) for Quinuas, and between 0.19 (pF 2.5) and 0.25 (pF 0, pF 0.5) for Laipuna. Differential evolution models trained on the Quinuas datasets correspond to mean $RMSE_E$ values ranging from 0.11 (pF 0) to 0.17 (pF 2.5). The Laipuna differential evolution models resulted in the same mean $RMSE_E$ values ranging from 0.15 (pF 2.5) to 0.28 (pF 0) Mean $R^2_E$ values resulting from grid search, are varying between 0.03 (pF 0) and 0.09 (pF 1.5) for Quinuas and between 0.03 (pF 0.5, pF 2.5) and 0.05 (pF 1.5) for Laipuna. The differential evolution algorithm resulted in mean $R^2_E$ values increasing from 0.58 (pF 2.5) to 0.79 (pF 0) for Quinuas and from 0.35 (pF 0.5) to 0.68 (pF 0) for Laipuna. As demonstrated by the scatterplots, the grid search models roughly reproduce the mean water retention values, while the models with parameter tuning by differential evolution are able to explain more of the observations' variance. The five grid search predictions for each observation (Fig. 8.1 and 9.1), cover a smaller range than the differential evolution predictions (Fig. 8.2 and 9.2). Especially the differential evolution results of the Laipuna pF 0 and pF 0.5 models are characterized by comparatively high variance. Caused by the better adjustment to the modeling problem, the differential evolution models show a higher predictive performance than the models tuned by grid search: mean $R^2_E$ values are up to 25 (Quinuas, pF 0) and 19 (Laipuna, pF 2.5) times higher, while scaled $RMSE_E$ values are up to 2.1 (Quinuas, pF 0) and 1.3 (Laipuna, pF 2.5) times lower than those obtained by grid search. This corresponds to the scatterplots (Fig. 8 and 9): the largest difference between grid search and differential evolution can be recognized for the pF 0 (Quinuas) and pF 2.5 (Laipuna) models. The higher variability of the differential evolution predictions corresponds to the differential evolution tuning parameter values covering a wider range than those achieved by applying grid search (Section 3.3). For Quinuas, the decreasing predictive performance with increasing pF values can probably be attributed to the lack of further predictors. While the predictors BD and SOC content are able to explain most of the water retention values at pF 0 to pF 1.5, the lack of predictors related to the soil matrix, e.g. PSD information, prevents further improvement for pF 2.5. In pedometrics, studies with a direct comparison of grid search and mathematical optimization applied for parameter tuning in machine learning are scarce. In fact we are only aware of one application: Wu et al. (2016) compared both tuning strategies to train support vector machine (SVM) models for the prediction of soil contamination in the Jiangxi Province (China). Their results are contradictory: Overall, using optimization to tune three SVM parameters led to the best model performance. Unfortunately, the comparison with grid search was only applied to a reduced two-parameter tuning problem. Surprisingly, here, grid search outperformed the tested optimization algorithms. Unfortunately, the tuning of a different number of SVM parameters is hampering the direct comparison. Still, the results of Wu et al. (2016) show that a lucky selection of predefined parameter vectors can result in grid search outperforming optimization algorithms – in particular, if the number of optimization iterations is small. Overall, the more values are tested during parameter tuning (grid

search or optimization), the higher the probability of finding the global optimum. Wu et al. (2016) did not mention the number of iterations of the optimization algorithms, but we assume that increasing the number of iterations would have led to results

that are at least as good as those achieved by grid search. Even though the benefits of optimization algorithms towards grid search are obvious, further direct comparisons of mathematical optimization algorithms and grid search applied for machine learning parameter tuning in soil related research questions are necessary.

Overall, the predictive power of all differential evolution based Quinuas models and the Laipuna pF 0 and 2.5 models are comparable to other studies. Botula et al. (2013) for example obtained R² values ranging from 0.32 to 0.68 (pF 0) and from

0.60 to 0.68 (pF 1.5) by using the k-nearest neighbor algorithm for soil data originating from the Lower Congo. Keshavarzi et al. (2010) used an artificial neural network to predict water retention at different pF values for soils from the Qazvin province in Iran. Haghverdi et al. (2012) used the same machine learning technique on soils from northeastern and northern Iran. While Keshavarzi et al. (2010) gained $R^2$ values of 0.77 (pF 2.5) and 0.72 (pF 4.2), Haghverdi et al. (2012) reached $R^2$ values ranging from 0.81 to 0.95. In general, we expect model performance to improve by removing extreme values in the model input or by

using larger datasets. Even though they were not identified as multivariate outliers, the low water retention values are underrepresented in the Quinuas dataset. According to Guio Blanco et al. (2018), these values are primarily observed in the lower part of the river valley and include measurements from mineral soils. Furthermore, it needs to be tested if different model algorithms are able to improve PTFs for both research areas.

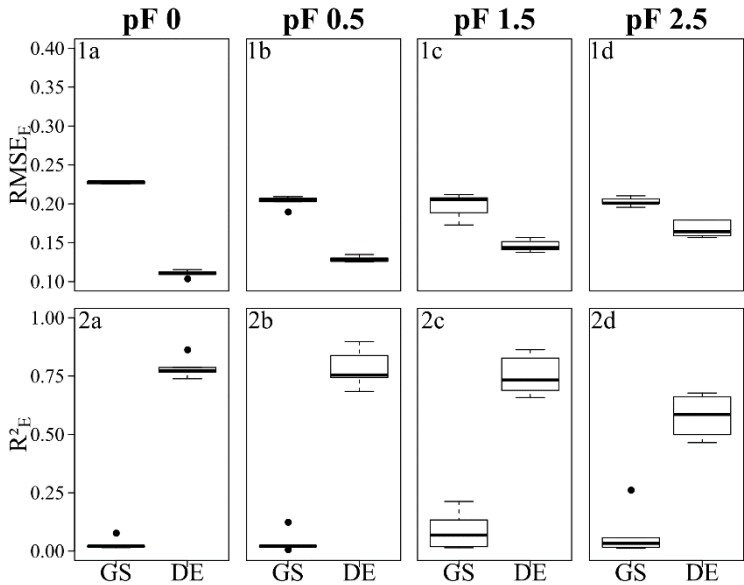

**Fig. 6. Error metrics of the Quinuas BRT models.** 6.1 RMSE$_E$, 6.2 R²$_E$, a) pF 0, b) pF 0.5, c) pF 1.5, d) pF 2.5. Each boxplot is based on five values resulting from five CV repetitions. GS = grid search, DE = differential evolution algorithm. Error metrics were calculated based on response variables scaled to the range [0, 1]

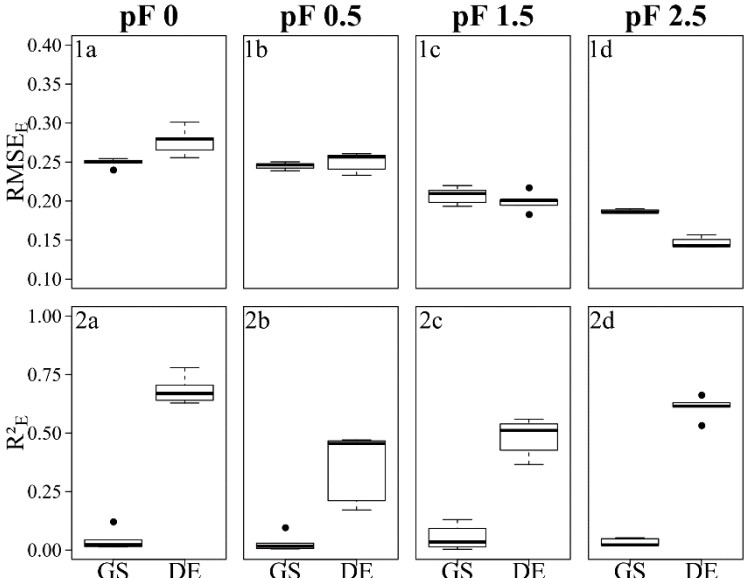

**Fig. 7. Error metrics of the Laipuna BRT models.** 7.1 $RMSE_E$, 7.2 $R^2_E$, a) pF 0, b) pF 0.5, c) pF 1.5, d) pF 2.5. Each boxplot is based on five values resulting from five CV repetitions. GS = grid search, DE = differential evolution algorithm. Error metrics were calculated based on response variables scaled to the range [0, 1].

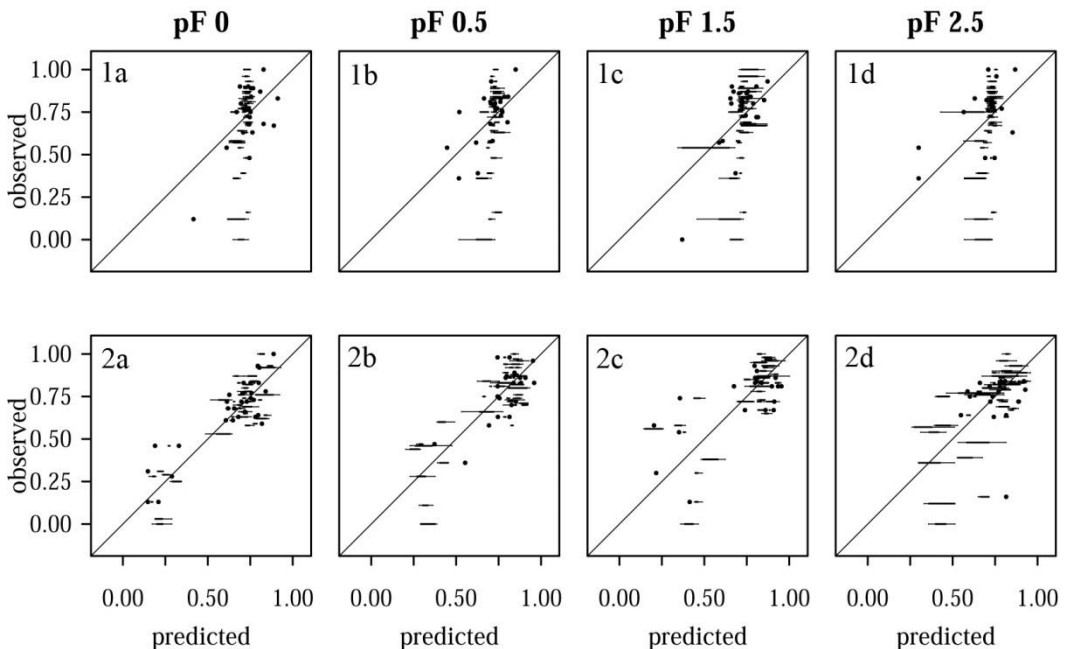

**Fig. 8. Comparison of predicted and observed water retention values for Quinuas.** 8.1 Models with tuning by grid search, 8.2 Models with parameter tuning by differential evolution, a) pF 0, b) pF 0.5, c) pF 1.5, d) pF 2.5. Each boxplot is based on five values resulting from five CV repetitions. Predicted and observed values were scaled to the range [0, 1].

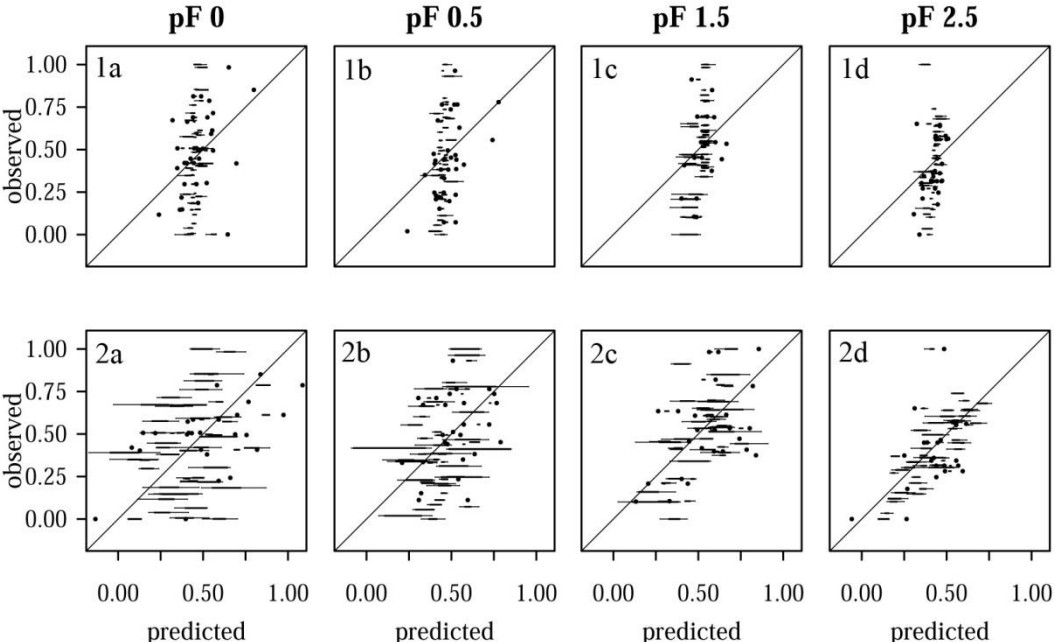

**Fig. 9. Comparison of predicted and observed water retention values for Laipuna.** 9.1 Models with tuning by grid search, 9.2 Models with parameter tuning by differential evolution, a) pF 0, b) pF 0.5, c) pF 1.5, d) pF 2.5. Each boxplot is based on five values resulting from five CV repetitions. Predicted and observed values were scaled to the range [0, 1].

### 3.3 Comparison to existing PTFs

Applying the existing PTFs with predictor variables sampled in Quinuas and Laipuna confirmed the good performance of the differential evolution BRT models. RMSE values of the respective PTFs are shown in Table 2. They were calculated by comparing the unscaled measured water retention of each soil profile to the water retention values calculated by applying the readily available PTFs. For Laipuna, mean $RMSE_E$ values of the differential evolution tuned BRT models were between 1.3 times (pF 2.5, Minasny and Hartemink (2011) and Tomasella et al. (2000)) and 9.3 times (pF 1.5, Barros et al. (2013)) better (Table 2). For Quinuas, the application of the differential evolution BRT models resulted in a mean $RMSE_E$ of 0.03, while applying the PTF of Boelter (1969) resulted only in an RMSE of 1.86. For BD higher than 370 kg m$^{-3}$ predictions became even negative. The high RMSE value is assumed to be caused by large differences between the temperate organic soils in Minnesota and the soils in Quinuas. This underlines the necessity of developing water retention PTFs specifically for tropical organic soils.

**Table 2. Unscaled root mean squared errors for the tested PTFs.** The best results for each matric potential are underlined. BRT PTF results are averaged.

| PTF [data origin, size of the dataset] | pF 0 | pF 0.5 | pF 1.5 | pF 2.5 |
|---|---|---|---|---|
| BRT PTF [Laipuna, 51] | 0.03 | 0.03 | 0.03 | 0.03 |
| Barros et al. (2013) [Brazil, 668] * | 0.18 | 0.12 | 0.28 | 0.07 |
| Gaiser et al. (2000) [Brazil and Niger, 627] | - | - | - | 0.06 |
| Minasny and Hartemink (2011) [various tropical regions, 652] | - | - | - | 0.04 |
| Nguyen et al. (2014) [Vietnam, 160] | - | - | 0.05 | 0.07 |
| Obalum and Obi (2012) [Nigeria, 54] | - | - | - | 0.15 |
| Pollacco (2008) [USA, 18552] | - | - | - | 0.07 |
| Tomasella et al. (2000) [Brazil, 630] * | 0.08 | 0.07 | 0.05 | 0.04 |

* applied to predict parameters of the Van Genuchten equation first.

### 3.4 Model parameters

The final tuning parameter values obtained by grid search and the differential evolution algorithm are summarized in Fig. 10 (Quinuas) and 11 (Laipuna). 625 previously defined parameter vectors were compared by grid search. On average 31 (pF 0, 0.5), 33 (pF 1.5) and 28 (pF 2.5) iterations of the differential evolution algorithm were necessary to find the optimal tuning parameter values for the Quinuas models. For Laipuna 32 (pF 0), 28 (pF 0.5), 25 (pF 1.5) and 22 (pF 2.5) iterations were needed. Differences between the parameter tuning techniques are most distinct for n.trees and shrinkage. Neglecting outliers, values obtained by the differential evolution algorithm cover a wider range than those resulting from grid search: while n.trees was in most cases set to the lowest tested value (100) by grid search, the differential evolution algorithm resulted in mean n.trees values (± standard deviation) ranging from $310 \pm 321$ (pF 0) to $810 \pm 1132$ (pF 1.5) for Quinuas and from $727 \pm 851$ (pF 0) to $1688 \pm 1345$ (pF 2.5) for Laipuna. Thereby, the mean n.trees values obtained by differential evolution parameter tuning are more than five (Quinuas) and more than ten times (Laipuna) higher than the mean grid search values. Neglecting extreme values, the shrinkage values resulting from the differential evolution algorithm also cover a wider range than the values obtained by the grid search tuning technique. For both areas, the shrinkage values were usually set to 0.001 or 0.01 by grid search, while applying the differential evolution algorithm resulted in mean shrinkage values ranging from $0.040 \pm 0.028$ (pF 0.5) to $0.047 \pm 0.030$ (pF 2.5) for Quinuas and from $0.034 \pm 0.03$ (pF 0) to $0.062 \pm 0.027$ (pF 2.5) for Laipuna. On average, the differential evolution shrinkage values are approximately 14 (Quinuas) and 17 (Laipuna) times higher than those obtained by grid search. The observed pattern is more complex for the other two tuning parameters: interaction depth and bag fraction. Although the selected parameter ranges differ for most pF values, the median interaction depth values are the same for half of the cases for grid search and tuning by the differential evolution algorithm. The median of the selected bag fraction is at the

upper limit for the Quinuas models that were tuned by the differential evolution algorithm, while grid search resulted in median

bag fraction values at the lower limit in two cases. Laipuna bag fraction values do not show this pronounced difference between grid search and tuning by the differential evolution algorithm.

The selected tuning parameter values correspond to the differential evolution based models having more predictive power than those adapted by the common grid search approach. Usually higher n.trees values, as received from the differential evolution algorithm, are known to improve model performance (Elith et al., 2008). However, according to the results of Elith et al.

(2008), by using more trees the shrinkage parameter gets smaller. The comparatively high differential evolution shrinkage values are an indication of the n.trees values still being too small. For both areas, the differential evolution values for n.trees and shrinkage, covering a wider range than the grid search results, are assumed to be caused by an incomplete optimization caused by not using enough iterations or the algorithm being stuck in a local optimum. This corresponds to the high prediction variability of the final differential evolution models derived for Laipuna (Fig. 9). It should be noted that model performance

depends on the combination of parameter values. However, as n.trees and shrinkage control how precisely the model learns the input data structure, these parameters are assumed to be more important than interaction depth and bag fraction. In this case, there would not even be an optimum for the latter two parameters. Especially for Laipuna, this explains the interaction depth and bag fraction values of both tuning techniques covering the whole range of possible values. The bag fraction differences between differential evolution and grid search tuning remain unexplained. For both parameter tuning techniques,

increasing the number of parameter values to be tested enhances the probability of finding the global optimum. For grid search, this can be realized by increasing the number of values to be compared for each tuning parameter. Increasing the number of iterations and starting with larger and thereby more heterogeneous initial populations is expected to do the same for differential evolution. This is assumed to result in less variable differential evolution results. However, for tuning continuous parameters, it is impossible to know the necessary number of iterations in advance. Accordingly, a trade-off between computing time and

the probability of finding the global optimum has to be made for any parameter tuning technique. Besides increasing the number of iterations and the number of initial vectors, the risk of the differential evolution algorithm getting stuck in a local optimum can also be reduced by changing the parameters "crossover probability" and the "mutation scaling factor" as well as applying another mutation strategy (Das and Suganthan, 2011). To overcome the problem of choosing the right control parameters as well as the mutation strategy, self-adaptive differential evolution algorithms (e.g. Nahvi et al., 2016; Pierezan et

al., 2017; Qin et al., 2009), which are able to automatically adjust their settings during the optimization process, could be applied in future studies. Furthermore, a larger model input of high quality would result in more explicit relationships between response and predictor variables that can be detected and reproduced more easily by the BRT models. This is assumed to reduce the probability of the differential evolution algorithm getting stuck into a local optimum as well as the number of required iterations. In general, the superiority of differential evolution needs to be verified by applying it to further machine

learning algorithms and applications, and by comparing it to further parameter tuning techniques.

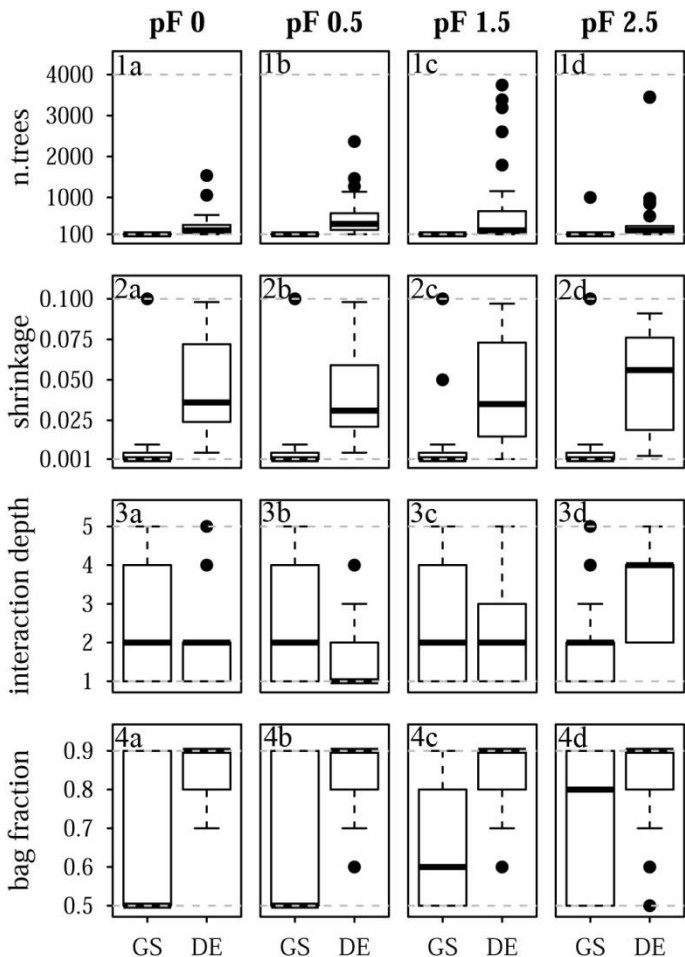

**Fig. 10. Selected tuning parameter values for Quinuas.** 10.1 n.trees, 10.2 shrinkage, 10.3 interaction depth, 10.4 bag fraction, a) pF 0, b) pF 0.5, c) pF 1.5, d) pF 2.5. Each boxplot is based on 25 values corresponding to the five-fold CV with five repetitions. Dashed grey lines are indicating the chosen optimization limits. GS = grid search, DE = differential evolution algorithm.


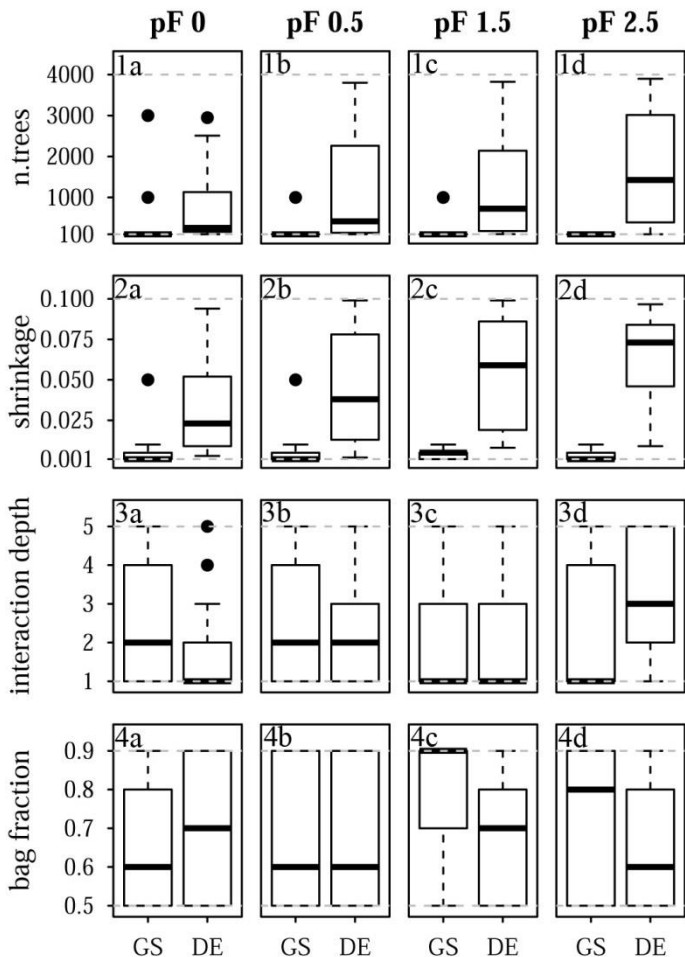

**Fig. 11. Selected tuning parameter values for Laipuna.** 11.1 n.trees, 11.2 shrinkage, 11.3 interaction depth, 11.4 bag fraction, a) pF 0, b) pF 0.5, c) pF 1.5, d) pF 2.5. Each boxplot is based on 25 values corresponding to the five-fold CV with five repetitions. Dashed grey lines are indicating the chosen optimization limits. GS = grid search, DE = differential evolution algorithm.

## 4 Conclusions

We successfully developed new PTFs for two tropical mountain regions. The comparison with readily available PTFs showed their high performance to predict soil water retention for the soils in these areas. This is of particular importance for soil hydrological modeling. Whether the two PTFs may also be applied to other areas of similar soils still has to be tested. The developed PTF for the Páramo area provides a novelty since PTFs for tropical organic soils under volcanic influence were unavailable until now.

Furthermore, our study presents the first successful application of parameter tuning by differential evolution in PTF development. The comparison with the standard grid search technique revealed the superiority of the differential evolution algorithm and emphasizes the importance of parameter tuning for the successful application of machine learning models. Of

course, this finding has to be confirmed by further applications in pedometrics including different machine learning algorithms. Especially for tuning of machine learning algorithms with continuous parameters, we hope to promote the implementation of optimization algorithms within the pedometrics community.

**Data availability**

The developed PTFs as well as the underlying datasets are available from https://doi.org/10.17605/OSF.IO/7UBWY (Ließ et al., 2020).

**Author contribution**

- Soil sampling and lab measurements (VBG, ML)
- Model setup and data analysis (AG, ML, ME)
- Manuscript writing (AG, ML, ME)
- Conceptual embedding (ML)

**Competing interests**

The authors declare that they have no conflict of interest.

**Acknowledgments**

This research was funded by the German Research Foundation (DFG) as part of the Platform for Biodiversity and Ecosystem Monitoring and Research in South Ecuador (PAK 825, LI 2360/1-1). Logistic support by the NGO Nature and Culture International (NCI) and the municipal public agency ETAPA is gratefully acknowledged.

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
