# Peer review of "Development of pedotransfer functions for water retention in tropical mountain soilscapes: Spotlight on parameter tuning in machine learning"

_SOIL, 2019_

## Short Comment (SC1) · 19 Nov 2019

I agree with the authors that a grid-search process is by far the most used method for parameter tuning in Pedometrics, but claiming that (L. 64) differential evolution has been applied for the first time in Gebauer et al. (2019) is not correct. Differential evolution is routinely used in Pedometrics since several years, in particular to find optimal values of parameters, see for example https://doi.org/10.1016/j.geoderma.2018.03.010 or https://doi.org/10.1016/j.catena.2016.02.016. For parameter tuning in ML applied to soil mapping, Wu et al. (2016) (10.1007/s11368-016-1374-9) compared a genetic algorithm, Particle Swarm optimization and a grid search process to find optimal ML tuning

parameters.

Without any surprise an optimization algorithm leads to more optimal parameter values than a grid-search process. This is obvious because a global optimization algorithm searches for any possible value within pre-defined boundaries while a grid-search is limited to a user-defined number of values. It should be noted that grid-search parameter tuning is by far the most used because the user knows in advance the number of iterations that will be needed. This is impossible to estimate with differential evolution, even though user-defined values in differential evolution can make the optimization to converge faster. This is a major limitation and the main reason why differential evolution (or any global optimization algorithm such as SA or PSO) are not routinely used for ML tuning parameter optimization.

Parameter tuning of ML models is computationally expensive and in most cases differential evolution will be too slow. In my experience differential evolution can need several hundreds to several thousands of iterations to find a global optimum.

For this reason, when ML parameters need to be tuned, other more efficient algorithms are used in the ML literature. Bayesian optimization is one of them. Bayesian optimization has been designed for parameter tuning of ML models but is much faster than other global optimization algorithms. Bayesian optimization finds the optimal tuning parameter values in very few iterations. Another advantage is that the algorithm does not need specific pre-defined boundaries. I personally applied it for ML parameters tuning in https://doi.org/10.5194/soil-5-107-2019.

Can the authors make a plot with in the x axis the number of iterations and in the y axis the value of the tuning parameters? This would be useful to see how the algorithm converges.

---

## Referee Comment (RC1) · Anonymous Referee #1 · 19 Dec 2019

Summary

In the manuscript by Gebauer et al. new pedotransfer functions are derived to predict the soil water retention at pF 0, 0.5, 1.5 and 2.5 for organic soils with volcanic influence and tropical mineral soils. The soil samples were collected in two tropical mountain regions in Ecuador. The authors use boosted regression trees method to predict the soil water retention from easily available soil properties. They analyse the performance of two parameter tuning methods: 1. the widely used grid search and 2. differential evolution optimization. The performance of the newly derived PTFs are compared to existing tropical PTFs on one studied site. The authors found that 1. the differential

evolution method outperformed the grid search, 2. the newly derived PTFs were more accurate than the readily available ones.

General comments

The authors derived pedotransfer functions also for soils which rarely have information on soil hydraulic properties: organic soils under volcanic influence with low bulk density and high organic carbon content. Soil hydraulic behaviour of these soils are unique, pedotransfer functions derived on mineral, non volcanic soils cannot be successfully applied for them. The presented study fills a gap related to describing soil hydraulic properties of an underrepresented soil system. For those, who are non-experts in the topic of the manuscript, it would be important to explain why separate pedotransfer functions were developed for the two studied sites.

The manuscript is well structured, the methods used to derive and optimize the predictions are adequate. Machine learning methods are common tools to predict soil properties. The selection of method to optimize the parameters of a particular machine learning method depends on the size of the dataset (number of samples in the training dataset), number of predictors, type of algorithm (how many parameters have to be optimized) and computation capacity. It would be important to mention and discuss these factors when performance of grid search and differential evolution algorithm is compared.

It is very positive that the newly derived PTFs are compared with existing PTFs – derived for tropical soils and available from the literature –, but it is not clear how many samples of the Laipuna site were used for it. The number of samples used for the analysis are generally not clear in the text, suggestions for clarification are included in the specific comments below.

It would be more informative to show results based on unscaled values, i.e.: Figures 6-8 and lines 186-191 and 219-223. It would enhance comparison of the results with the literature.

[Figure]

Authors could put more focus on soil physical interpretation of the results. Prediction of soil hydraulic properties of such specific soils as presented in the study is particular and very useful. The novelty of the paper could be connected to this.

It is not mentioned how the derived pedotransfer functions can be accessed.

Specific comments

L1-2: please specify in the title somehow that you derived PTFs to predict soil hydrological properties.

L15: please add name of the country.

L82, 88: could you please add the WRB (IUSS Working Group WRB: World Reference Base for Soil Resources 2014. International soil classification system for naming soils and creating legends for soil maps, Rome, 121 pp., 2014.) name of the most typical soils occurring in the studied sites?

L98-99: please give number of soil profiles and soil samples, instead of number of sites and sampling depth.

L99: Please add e.g. suction applied in hPa or matric potential head in cm. Why did you choose to measure water retention at pF 0, 0.5, 1.5 and 2.5? What is the reason for not determining water retention at pF 4.2?

L100: please add reference for the determination of soil water retention and BD.

L101-103: please add if it is the standard method in Germany for the determination of PSD.

L104: if I have understood it well, you didn't measure PSD for the Quinuas site because of the high SOC content. Please mention it here and shortly describe the reason for it.

L105: please add reference to CaCO3 determination.

L126: it would be helpful to shortly summarize what happens in 1) grid search and 2)

with the differential evolution algorithm.

L129: ... were compared in grid search... Please add the name of R package you used to apply the grid search for tuning the parameters.

L134: please define here the meaning of v. Is the meaning of v = 100 the same in L139?

L147: add somewhere in the materials and methods section which soil variables you use as predictors by sites. In the present manuscript reader gets information about it only from Fig. 4. under Results and discussion section.

L177: please add that the description is in the text e.g.: ... to the modelling steps described in text ...

L182: what do you mean that number of samples was 51 and 46? Please rephrase the sentence accordingly.

L186-191: please add unscaled RMSE value with unit as well already here, because readers are familiar with that.

L199-201: The two sentence could be concatenated: the one starting with "Measured BD ..." and the other starting with "The water ...".

L203: please explain what you mean by "correspond to soil samples with a higher proportion of mineral components or andic properties".

L208: you could highlight here why it is an interesting dataset for deriving a new hydraulic pedotransfer function.

L210: Figure 4: - it would help comparison of Quinuas and Laipuna data if the min. and max. values of y-axis would be the same, you could include violin plot of both sites in one plot: one plot for OC and another for BD, - add in caption that PSD of Quinuas was not measured, and shortly add reason for it, - instead of showing the cumulative distribution of the PSD (Fig 4. c)) texture triangle diagrams separately would be more

informative, - please add number of samples to the figure, e.g: in title or caption.

L213: Figure 5: - please add number of samples to the figure, e.g: in title or caption.

L218: you didn't mention in Materials and methods that you use scaled water retention values in the algorithm, please add it there and the reason for it there.

L219-223: please add unscaled RMSE value with unit as well.

L232: . . . models, regarding RMSEE and RE values . . .

L234: please consider to delete "However,".

L237-238: please consider if number of samples can influence the performance of parameter tuning in sentence starting with "Probably". Maybe it could be discussed how performance of tuning methods would change if you could include other predictors as well, e.g.: pH, CEC, etc.

L242-243: please mention under materials and methods the mean stone content of the Laipuna samples, if stoniness is characteristic for those.

L245: It is not clear what predictors were used to predict water retention of Quinuas. Please add it as mentioned before. It could be explained which suction heads can be covered by the predictors you have for Quinuas. Sentence starting with "PSD" should be moved under Materials and methods section, please see previous comments.

L246: Why performance of Laipuna PTFs for pF0, pF 0.5 and pF 1.5 is lower that that of Quinuas? Please discuss how those could be improved.

L253-273: Please add title to that section, to highlight that you applied existing PTFs on the sites to compare the performance of the newly derived PTFs to those.

L253: please add number of samples of the Laipuna dataset. Did you use the test set for the comparision?

L254: . . . PTFs from the literature were selected . . . Or add something similar.

L254-256: please move it under Materials and methods.

L256: it might be more precise to write that silt and sand content was converted to 2-50 $\mu$m and 50-2000 $\mu$m fractions by spline interpolation to calculate the USDA texture classes.

L263: add unit of RMSE.

L266-267: please mention other factors as well which could increase the performance of the PTFs.

L271: Table 2: - add number of samples – by pF values – used to test the newly derived and existing PTFs, did you use the test set of Laipuna dataset? - please use also here pF 0, 0.5, 1.5 and 2.5 instead of Theta 0, 0.5, 1.5 and 2.5., - add unit of the RMSE.

L275-278: for easier comparison Figures 6 and 7 could be concatenated by using grouped boxplots.

L280-285: based on Figure 5 observed pF values of Quinuas site is greater then 0.30 cm3/cm3, for Laipuna those are greater than 0.20 cm3/cm3. Please check in calculations why you have observed pF values close to 0 cm3/cm3 on Figures 8 and 9. Or are those scaled observed and scaled predicted variables? It would be more informative to show the scatterplot for not scaled observed and predicted values. Please revise Figures 8 and 9.

L293-294: Sentence starting with "Difference": there is difference between GS and DE in case of the bag fraction as well. Is it possible to show which parameter – among number of trees, shrinkage, interaction depth, bag fraction – has the most dominant influence on the performance of BRT?

L318: . . . for the final differential evolution models derived for Laipuna site (Fig. 9) . . .

L334, 339: In the caption of Figure 10 and 11: add number of tested parameter vectors for both method.

L343-344: please note that in most of the cases local PTFs perform better than PTFs trained on dataset originating from elsewhere with different soil forming factors. Please revise the sentence.

L351-354: please consider to concatenate the last two sentence of the conclusions to better balance highlight both on the newly derived PTFs and results of comparing parameter tuning methods.

L354: please consider to provide availability of derived PTFs – which you recommend to use – for users.

Technical comments

L67: Please add the country after Páramo.

L151: The acronym of BRT is not included in the flowchart.

L343: . . . readily available . . .

---

## Referee Comment (RC2) · Anonymous Referee #2 · 14 Jan 2020

The objective of the study is to develop pedotransfer function for water contents at four pressure heads (PF 0, PF 0.5, PF 1.5, PF 2.5) for two tropical mountain regions with high soil organic carbon content. Boosted regression tree technique was used to fit the models for both areas considering two tuning procedures to determine the regression tree-model parameters (n.tree, shrinkage, interation depth, bag fraction): grid search and differential evolution, the latter showing better results on the water retention estimates for the both areas. The work also compared the performance of the proposed PTFs with other PTFs from literature confirming the better performance of the proposed models. I congratulate the authors for the effort in collecting physico-chemical and hydrological data in such atypical soils and for using innovative techniques, such

as the differential evolution, in order to get better results on the models fits. I also congratulate them for developing PTFs for organic soils which are not so common in the literature. The work in well written and structured and the subject is well posed Some general and specific comments are summarized below: a) Line 45: In organic Finnland soils? b) It was not clear in the text why you have chosen the boosted regression tree models (Lines 72-73); c) The sentence in line 97-98 should be reformulated ("It allows representing a research. . ..to the accessible area"). The way it was written was unclear to me. d) Line 105. Sometimes outliers should carry important information from the studied area. You should detail the reason of removing them. e) The description of the boosted regression tree should be improved by describing clearer its fitting procedure (Lines 110-115). f) Line 144: What the word "respectively" is related to? g) Line 182: After explaining the reasons for excluding the outliers it would be interesting to inform the range of their values for each soil property; h) Lines 182-190: What is right and left- skewed distribution? It is not clear. i) Line 199: I suggest to correct this sentence: "..organic matter being characterized by a high water holding capacity" to *organic matter which is associated with soils with a high water holding capacity* j) Line 218: How the scaled water retention value was defined? This needs to be clarified; k) Line 230: Change Fig.11 and 12 to Fig 8 and 9; l) Line 240: Change Section 3.2 to Section 3.3; m) Line 245: "PSD measurements were not included..in this area". This sentence should go to Section 3.1 when you call Fig.4 in the text. n) Line 234: Change Fig.9 a-d and 10 a-d to Fig.8 a-d to Fig.9 a-d; o) Lines 253-263: • Did you apply the PTFs from the literature to the Laipura soils considering their range values applicability? • The test Laipura soils were included in the calibration of the proposed PTFs (BRT PTF – Table 2)? This need to be clarified. p) Avoid vague sentences: ex: "these values are.."(Line 268), "in this case"(Line 320), "this might.." (line 320), "This might also result" (lines 325-326); q) The code of the proposed models should be presented; r) Is it possible to provide the study database to the readers?

---

## Author Comment (AC1) · 10 Feb 2020

Dear Mr. Wadoux,

Thank you for your time and valuable comments.
Please find our replies below each comment. They are displayed in blue. Line numbers in the replies refer to the revised manuscript with tracked changes. Line numbers in the reviewer comments refer to the first manuscript version.

Kind regards,
Anika Gebauer
* * *
I agree with the authors that a grid-search process is by far the most used method for parameter tuning in Pedometrics, but claiming that (L. 64) differential evolution has been applied for the first time in Gebauer et al. (2019) is not correct. Differential evolution is routinely used in Pedometrics since several years, in particular to find optimal values of parameters, see for example https://doi.org/10.1016/j.geoderma.2018.03.010 or https://doi.org/10.1016/j.catena.2016.02.016 .
Further applications of the differential evolution algorithm in soil-related research questions were added to lines 70 ff.. Nevertheless, we would like to emphasize that applications for machine learning parameter tuning in pedometrics are scarce.

For parameter tuning in ML applied to soil mapping,Wu et al. (2016) (10.1007/s11368-016-1374-9) compared a genetic algorithm, Particle Swarm optimization and a grid search process to find optimal ML tuning parameters.
Without any surprise an optimization algorithm leads to more optimal parameter values than a grid-search process. This is obvious because a global optimization algorithm searches for any possible value within pre-defined boundaries while a grid-search is limited to a user-defined number of values.
The results of Wu et al. (2016) show that mathematical optimization algorithms do not necessarily outperform grid search: tuning two real-valued parameters using grid search led to better results than tuning the same parameters applying optimization algorithms. This shows that a lucky selection of predefined parameter combinations to be tested can result in grid search outperforming optimization algorithms – in particular, if the number of optimization iterations is small. The more values are tested during parameter tuning, the higher the probability of finding the global optimum. According to Wu et al. (2016), the better performance of grid search was caused by grid search exhaustively evaluating every predefined parameter value, while optimization algorithms can get stuck in local optima. Wu et al. (2016) did not mention the number of iterations of the optimization algorithms, but we assume that increasing the number of iterations would have led to results that are at least as good as those achieved by grid search.
Tuning three parameters with the genetic algorithm resulted in a better model performance than tuning two parameters with grid search, the genetic algorithm or the particle swarm optimization (Wu et al. (2016), Fig. 2). The different number of parameters hampers the direct comparison of the applied tuning techniques. The difference in model performance is most likely caused by tuning the third parameter and not by the parameter tuning technique itself.
Even though the benefits of optimization algorithms towards grid search are obvious, further direct comparisons of mathematical optimization algorithms and grid search applied for machine learning parameter tuning in soil related research questions are necessary. We compared the differential evolution algorithm to grid search because it is superior to various optimization algorithms including particle swarm optimization and genetic algorithms (lines 62 ff.).

It should be noted that grid-search parameter tuning is by far the most used because the user knows in advance the number of iterations that will be needed.
This is impossible to estimate with differential evolution, even though user-defined values in differential evolution can make the optimization to converge faster. This is a major limitation and the main reason why differential evolution (or any global optimization algorithm such as SA or PSO) are

not routinely used for ML tuning parameter optimization.

Parameter tuning of ML models is computationally expensive and in most cases differential evolution will be too slow. In my experience differential evolution can need several hundreds to several thousands of iterations to find a global optimum.

We agree that it is beneficial to know the number of tuning iterations in advance. However, the differential evolution algorithm allows defining a stopping criterion. In this case, we stopped the differential evolution optimization either after 10 iterations without improvement or a maximum number of 200 iterations (lines 183 f.).

Regardless of the chosen parameter tuning technique, restricting the number of parameter values to be tested reduces the probability of finding the global optimum. For tuning real-valued parameters, it is impossible to know the necessary number of iterations in advance. A trade-off between computing time and the probability of finding the global optimum has to be made for any parameter tuning technique.

We agree that parameter tuning of machine learning models requires computing power. But with computers becoming more efficient and the possibility of parallelization, a parameter tuning technique should be judged based on the predictive power of the resulting machine learning model and not on the required computing resources.

It requires more computational resources to find the global optimum of real-valued parameters by testing every possible parameter combination during grid search instead of converging towards the optimum using a well-adjusted optimization algorithm. Making the wrong decisions in predefining the parameter values to be tested might even prohibit grid search from ever reaching the global optimum in tuning real-valued parameters.

For this reason, when ML parameters need to be tuned, other more efficient algorithms are used in the ML literature. Bayesian optimization is one of them. Bayesian optimization has been designed for parameter tuning of ML models but is much faster than other global optimization algorithms. Bayesian optimization finds the optimal tuning parameter values in very few iterations. Another advantage is that the algorithm does not need specific pre-defined boundaries. I personally applied it for ML parameters tuning in https://doi.org/10.5194/soil-5-107-2019 .

Thank you for recommending Bayesian optimization. We see the advantages of the proposed parameter tuning technique. However, comparisons of both algorithms led to contradictory results: while some studies found Bayesian approaches to be superior (e.g. Carr et al. 2016*) others reported the opposite result (e.g. Schmidt et al. 2019**;  Salt & Howard 2015***). Further comparisons of Bayesian optimization and differential evolution in machine learning parameter tuning applications are necessary concerning computing requirements and the resulting model performance.

* Carr, Garnett and Lo. "BASC: Applying Bayesian Optimization to the Search for Global Minima on Potential Energy Surfaces." ICML (2016).

** Schmidt, Safarani, Gastinger, Jacobs, Nicolas and Schülke, "On the Performance of Differential Evolution for Hyperparameter Tuning". *International Joint Conference on Neural Networks (IJCNN)*, Budapest, Hungary, 2019, pp. 1-8. (2019) doi: 10.1109/IJCNN.2019.8851978

*** Salt, Howard, Indivera, Sandamirskaya, "Differential Evolution and Bayesian Optimisation for Hyper-Parameter Selection in Mixed-Signal Neuromorphic Circuits Applied to UAV Obstacle Avoidance". CoRR (2015). https://arxiv.org/abs/1704.04853

Can the authors make a plot with in the x axis the number of iterations and in the y axis the value of the tuning parameters? This would be useful to see how the algorithm converges.

In accordance with comment C42 of the anonymous referee #1, the number of differential evolution iterations was added to lines 397 ff. The suggested plot was not drawn, as a figure showing the converging algorithm for each tuning parameter and each cross validation fold would be rather confusing.

---

## Author Response (AR1)

Dear Mr. Vanderborght,

Thank you for receiving our revised manuscript. Please find the replies to your comments and to the reviewer comments below. Replies are displayed in blue and include the relevant changes made in the manuscript. Line numbers in the replies refer to the revised manuscript with tracked changes. Line numbers in the reviewer comments refer to the first manuscript version. Please find the marked-up manuscript version at the end of this .pdf file.

Kind regards
Anika Gebauer
* * *
**Editor comments to the author**

I agree with your responses to the reviewer comments on your paper. Your paper is mainly about a methodology to determine a pedotransfer function using a boosted regression tree and how the parameters of this model could be derived (using grid search or a differential evolution algorithm). In view of this, I think it is very important to address the comments by Alexandre Wadoux and include your responses in your paper.
The comments of Alexandre Wadoux were addressed and our responses were included in the paper (please see the replies to the comments of Alexandre Wadoux included in this .pdf file).

I would propose to put less focus on the derived pedotransfer functions themselves but more on the procedure on how to derive pedotransfer functions. The title of your paper puts the focus on these pedotransfer functions for tropical mountain soilscapes.
Our paper had two equally important aims (lines 80 ff.): developing water retention pedotransfer functions for two tropical soil-landscapes and comparing the differential evolution algorithm to grid search for parameter tuning. The title essentially refers to these two aspects.

But, would your pedotransfer functions be applicable for any other tropical mountain soilscape than the two that you considered? Since you developed two different pedotransfer functions, one for each landscape, I really doubt this. In the paper, these transfer functions are rather black boxes and it is far from clear whether they have any predictive power for other sites, land uses, …, than the sites where they were developed for. They do not provide insights in which input variables are most important to explain the output variables (at least, that was not reported in the paper) and it is not clear how they could be applied to datasets with different input data.
Pedotransfer functions were developed for each soilscape separately, as both areas are characterized by soils with very different properties. We made these differences more explicit in the abstract (lines 11ff). Please also compare text sections in lines 94 ff. and 247 ff.:

- lines 11 ff.: "(…) two remote tropical mountain regions of rather different soil-landscapes, dominated by (1) peat soils and soils under volcanic influence with high organic matter contents, and (2) tropical mineral soils."
- lines 94 ff.: "Quinuas (…) the mean annual temperature ranges between 5.3 and 8.7 °C (…) mean annual precipitation varies between 900 and 1600 mm (…). The Quinuas catchment can be allocated to the Páramo ecosystem (…). The Laipuna dry forest region (…) mean annual temperature varies between 16 and 23 °C and mean annual precipitation between 540 and 630 mm (…)."
- lines 243 ff.: "BD values in Quinuas range from 64 to 807 kg m$^{-3}$, while SOC values vary between 8.8 and 46.4 wt.%. … water retention ranges from 0.25 (pF 2.5) to 0.94 cm$^3$ cm$^{-3}$ (pF

0). (...) For Laipuna, BD ranges between 1157 and 1727 kg m$^{-3}$(...) SOC content (...) varies between 0.4 and 3.8 wt. %. (...) Water retention values are ranging between 0.25 (pF 2.5) and 0.61 cm$^3$ cm$^{-3}$ (pF 0). (...) Quinuas soils go along with the low density, porous soils, rich in organic material...Comparatively high water retention values can be attributed to the porous structure of Páramo soils (...)".

We are well aware that the collected dataset used for PTF development is rather small. Accordingly, applicability of the PTFs to other tropical soilscapes is not necessarily given but has to be tested. However, currently, there is a lack of PTFs for tropical soilscapes (compare lines 36-49, 223-233). For tropical peat soils under volcanic influence, there is probably none at all (lines 82 f.).

You are going to make the pedotransferfunctions that you developed available. It is however also necessary to make the dataset with input and output data available. This is important for the further improvement and generalization of pedotransfer functions.

The dataset used for PTF development comprising response (output) variables and predictor (input) variables will be uploaded to the Open Science Framework (OSF) upon acceptance (added to lines 488 ff.).

**Short comment of Alexandre Wadoux**

I agree with the authors that a grid-search process is by far the most used method for parameter tuning in Pedometrics, but claiming that (L. 64) differential evolution has been applied for the first time in Gebauer et al. (2019) is not correct. Differential evolution is routinely used in Pedometrics since several years, in particular to find optimal values of parameters, see for example https://doi.org/10.1016/j.geoderma.2018.03.010 or https://doi.org/10.1016/j.catena.2016.02.016 .

Thank you for pointing out these interesting manuscripts to us. Further applications of the differential evolution algorithm in soil-related research questions were added to lines 70 ff.. Nevertheless, we would like to emphasize that applications for parameter tuning in machine learning in pedometrics are scarce.

For parameter tuning in ML applied to soil mapping, Wu et al. (2016) (10.1007/s11368-016-1374-9) compared a genetic algorithm, Particle Swarm optimization and a grid search process to find optimal ML tuning parameters.
Without any surprise an optimization algorithm leads to more optimal parameter values than a grid-search process. This is obvious because a global optimization algorithm searches for any possible value within pre-defined boundaries while a grid-search is limited to a user-defined number of values.

The results of Wu et al. (2016) are contradictory. Please compare Lines 318 ff.: "In pedometrics, studies with direct comparison of grid search and mathematical optimization applied to parameter tuning in machine learning are scarce. Wu et al. (2016) compared both tuning strategies to train support vector machine (SVM) models for the prediction of soil contamination in the Jiangxi Province (China). However, their results are contradictory: In tuning only two real-valued SVM parameters, grid search outperformed the tested optimization algorithms, while tuning three parameters led to the opposite result. This shows that a lucky selection of predefined parameter vectors can result in grid search outperforming optimization algorithms – in particular, if the number of optimization iterations is small. Overall, the more values are tested during parameter tuning (grid search or optimization), the higher the probability of finding the global optimum. Wu et al. (2016) did not mention the number of iterations of the optimization algorithms, but we assume that increasing the number of iterations would have led to results that are at least as good as those achieved by grid search. Even though the benefits of optimization algorithms towards grid search are obvious, further direct comparisons of mathematical optimization algorithms and grid search applied for machine learning parameter tuning in soil related research questions are necessary."

It should be noted that grid-search parameter tuning is by far the most used because the user knows in advance the number of iterations that will be needed.
This is impossible to estimate with differential evolution, even though user-defined values in differential evolution can make the optimization to converge faster. This is a major limitation and the main reason why differential evolution (or any global optimization algorithm such as SA or PSO) are not routinely used for ML tuning parameter optimization.
Parameter tuning of ML models is computationally expensive and in most cases differential evolution will be too slow. In my experience differential evolution can need several hundreds to several thousands of iterations to find a global optimum.

We agree that it is beneficial to know the number of tuning iterations in advance. However, the differential evolution algorithm allows defining a stopping criterion. In this case, we stopped the differential evolution optimization either after 10 iterations without improvement or a maximum number of 200 iterations (lines 186 f.).
Regardless of the chosen parameter tuning technique, restricting the number of parameter values to be tested reduces the probability of finding the global optimum. Please see lines 447 ff.: "For both parameter tuning techniques, increasing the number of parameter values to be tested enhances the probability of finding the global optimum. (…) However, for tuning real-valued parameters, it is impossible to know the necessary number of iterations in advance. Accordingly, a trade-off between computing time and the probability of finding the global optimum has to be made for any parameter

tuning technique."

We agree that parameter tuning of machine learning models requires computing power. But with computers becoming more efficient and the possibility of parallelization, a parameter tuning technique should be judged based on the predictive power of the resulting machine learning model and not on the required computing resources.

We assume that it requires more computational resources to find the global optimum of real-valued parameters by testing every possible parameter combination during grid search instead of converging towards the optimum using a well-adjusted optimization algorithm. Making the wrong decisions in predefining the parameter values to be tested might even prohibit grid search from ever reaching the global optimum in tuning real-valued parameters (lines 55 ff.).

For this reason, when ML parameters need to be tuned, other more efficient algorithms are used in the ML literature. Bayesian optimization is one of them. Bayesian optimization has been designed for parameter tuning of ML models but is much faster than other global optimization algorithms. Bayesian optimization finds the optimal tuning parameter values in very few iterations. Another advantage is that the algorithm does not need specific pre-defined boundaries. I personally applied it for ML parameters tuning in https://doi.org/10.5194/soil-5-107-2019 .

Thank you for recommending Bayesian optimization. We see the advantages of the proposed parameter tuning technique, but want to mention that the differential evolution algorithm is able to outperform it in certain applications.

Please see lines 67 ff.:"It is also able to outperform bayesian approaches in certain applications. Comparisons of both algorithms led to contradictory results: while some studies found Bayesian approaches to be superior (e.g. Carr et al., 2016) others reported the opposite result (e.g. Schmidt et al., 2019)."

Further comparisons of other optimization algorithms including bayesian approaches and differential evolution in machine learning parameter tuning applications are necessary (added to lines 362 ff.).

Can the authors make a plot with in the x axis the number of iterations and in the y axis the value of the tuning parameters? This would be useful to see how the algorithm converges.

In accordance with comment C42 of the anonymous referee #1, the number of differential evolution iterations was added to lines 413 ff. We refrain from adding a plot, as a figure showing the converging algorithm for each tuning parameter and each cross validation fold would be rather confusing.

**Referee comment 1** (a number was added to each comment)

**General comments**

The authors derived pedotransfer functions also for soils which rarely have information on soil hydraulic properties: organic soils under volcanic influence with low bulk density and high organic carbon content. Soil hydraulic behaviour of these soils are unique, pedotransfer functions derived on mineral, non volcanic soils cannot be successfully applied for them. The presented study fills a gap related to describing soil hydraulic properties of an underrepresented soil system. For those, who are non-experts in the topic of the manuscript, it would be important to explain why separate pedotransfer functions were developed for the two studied sites.

The manuscript is well structured, the methods used to derive and optimize the predictions are adequate. Machine learning methods are common tools to predict soil properties. The selection of method to optimize the parameters of a particular machine learning method depends on the size of the dataset (number of samples in the training dataset), number of predictors, type of algorithm (how many parameters have to be optimized) and computation capacity. It would be important to mention and discuss these factors when performance of grid search and differential evolution algorithm is compared.

It is very positive that the newly derived PTFs are compared with existing PTFs – derived for tropical soils and available from the literature –, but it is not clear how many samples of the Laipuna site were used for it. The number of samples used for the analysis are generally not clear in the text, suggestions for clarification are included in the specific comments below.

It would be more informative to show results based on unscaled values, i.e.: Figures 6-8 and lines 186-191 and 219-223. It would enhance comparison of the results with the literature.

Authors could put more focus on soil physical interpretation of the results. Prediction of soil hydraulic properties of such specific soils as presented in the study is particular and very useful. The novelty of the paper could be connected to this. It is not mentioned how the derived pedotransfer functions can be accessed.

Thank you for your general comments. Our answers can be found below, next to the corresponding specific comments.

**Specific comments**

**C1:** L1-2: please specify in the title somehow that you derived PTFs to predict soil hydrological properties.

We changed the title to "Development of pedotransfer functions for water retention in tropical mountain soilscapes: Spotlight on parameter tuning in machine learning"

**C2:** L15: please add name of the country.

The name of the country was added (line 16).

**C3:** L82, 88: could you please add the WRB (IUSS Working Group WRB: World Reference Base for Soil Resources 2014. International soil classification system for naming soils and creating legends for soil maps, Rome, 121 pp., 2014.) name of the most typical soils occurring in the studied sites?

Due to monetary short cuts in the project grants we do not have all the required laboratory data in order to classify the soil profiles according to WRB. This is why we refrain from mentioning specific soil reference groups.

**C4:** L98-99: please give number of soil profiles and soil samples, instead of number of sites

and sampling depth.
Lines 118 ff. were adapted.

**C5:** L99: Please add e.g. suction applied in hPa or matric potential head in cm. Why did you choose to measure water retention at pF 0, 0.5, 1.5 and 2.5? What is the reason for not determining water retention at pF 4.2?
The suction applied in hPa was added to lines 121 f.. Water retention at pF 4.2 was not determined because of high soil organic matter contents in Quinuas (added to lines 124 f.). To be able to compare the model input of both research areas, water retention was measured at the same pF values.

**C6:** L100: please add reference for the determination of soil water retention and BD.
References were added in lines 122 f. and 126.

**C7:** L101-103: please add if it is the standard method in Germany for the determination of PSD.
There are several methods to determine soil texture. We refrain from calling one of them a standard method. The applied methodology is given in lines 129 ff.

**C8:** L104: if I have understood it well, you didn't measure PSD for the Quinuas site because of the high SOC content. Please mention it here and shortly describe the reason for it.
Please refer to lines 132 f. and lines 98 f.

**C9:** L105: please add reference to CaCO3 determination.
Adapted accordingly in lines 127 f.

**C10:** L126: it would be helpful to shortly summarize what happens in 1) grid search and 2) with the differential evolution algorithm.
Lines 167 ff. were adapted accordingly. The differential evolution algorithm is explained in lines 172 ff.

**C11:** L129: … were compared in grid search… Please add the name of R package you used to apply the grid search for tuning the parameters.
We did not use an R package for grid search and programmed it on our own.

**C12:** L134: please define here the meaning of v. Is the meaning of v = 100 the same in L139?
Yes, adapted accordingly in line 177.

**C13:** L147: add somewhere in the materials and methods section which soil variables you use as predictors by sites. In the present manuscript reader gets information about it only from Fig. 4. under Results and discussion section.
Soil variables that were used as predictors were explained in Section 2.2 (added to lines 125 f.).

**C14:** L177: please add that the description is in the text e.g.: … to the modelling steps described in text …
The caption of Fig. 3 was adapted accordingly.

**C15:** L182: what do you mean that number of samples was 51 and 46? Please rephrase the sentence accordingly.
The sentences in lines 240 ff. were rephrased.

**C16:** L186-191: please add unscaled RMSE value with unit as well already here, because

readers are familiar with that.

Lines 186-191 (now lines 245 ff.) describe the dataset shown in Figure 4 and 5. No RMSE values are shown.

The BRT models were trained on scaled values, which is why we provided scaled RMSE values in Section 3.2. A text section to explain this aspect was added in lines 138 ff..

**C17:** L199-201: The two sentence could be concatenated: the one starting with "Measured BD …" and the other starting with "The water …".

The two sentences are separated as both refer to different literature references.

**C18:** L203: please explain what you mean by "correspond to soil samples with a higher proportion of mineral components or andic properties".

The sentence in lines 264 ff. was rephrased.

**C19:** L208: you could highlight here why it is an interesting dataset for deriving a new hydraulic pedotransfer function.

Please see lines 35 ff. and lines 80 ff.

**C20:** L210: Figure 4: - it would help comparison of Quinuas and Laipuna data if the min. and max. values of y-axis would be the same, you could include violin plot of both sites in one plot: one plot for OC and another for BD,
- add in caption that PSD of Quinuas was not measured, and shortly add reason for it,
- instead of showing the cumulative distribution of the PSD (Fig 4. c)) texture triangle diagrams separately would be more informative,
- please add number of samples to the figure, e.g: in title or caption.

The violin plots of both sides were not included into one plot because BD and SOC ranges are rather different for Laipuna and Quinuas.

The cumulative PSD distribution was used instead of separate texture triangle diagrams, as it allows to show mean values and standard deviations per texture class (added to the caption of Fig. 4). The caption of Fig. 4 was adapted to explain why PSD was not measured for Quinuas and to add the number of samples.

**C21:** L213: Figure 5: - please add number of samples to the figure, e.g: in title or caption.

The number of samples was added to the caption of Fig. 5.

**C22:** L218: you didn't mention in Materials and methods that you use scaled water retention values in the algorithm, please add it there and the reason for it there.

Please see comment C16.

**C23:** L219-223: please add unscaled RMSE value with unit as well.

Please see comment C16.

**C24:** L232: … models, regarding $RMSE_E$ and $R_E$ values …

The sentence relates to the difference between the scatterplots in Fig. 8 and 9.

**C25:** L234: please consider to delete "However,".

"However" was deleted (line 295).

**C26:** L237-238: please consider if number of samples can influence the performance of parameter tuning in sentence starting with "Probably". Maybe it could be discussed how performance of tuning methods would change if you could include other predictors

as well, e.g.: pH, CEC, etc.

The result of every tuning method (i.e. the parameter values) indirectly depends on the dataset, as parameter tuning means adjusting a model to the specific modelling problem / dataset (lines 52 f.). As discussed in e.g. lines 315 ff. or lines 339 ff. the number of samples and the number of predictors (and their information content) influence the performance of the BRT models. In general, a larger dataset of high quality results in more explicit relationships between response and predictor variables that can be detected and reproduced more easily by a model. This might reduce the required differential optimum iterations and the probability of getting stuck into a local optimum (added to lines 460 ff.).

Concerning the general questions:

The optimization methods should be chosen based on the type of model algorithm. Models with real valued parameters require a different tuning technique than those with discrete valued parameters (lines 55 ff.). The selection of the tuning technique does not depend on the number of parameters to be tuned. Grid search and optimization algorithms allow the tuning of only one parameter as well as the tuning of several parameters.

The computation capacity is still a limiting factor when it comes to optimization. But with computers becoming more efficient and the possibility of parallelization, a parameter tuning technique should be judged based on the predictive power of the resulting machine learning model and not on the required computing resources.

**C27:** L242-243: please mention under materials and methods the mean stone content of the Laipuna samples, if stoniness is characteristic for those.

The stone content was not determined for all samples.

**C28:** L245: It is not clear what predictors were used to predict water retention of Quinuas. Please add it as mentioned before. It could be explained which suction heads can be covered by the predictors you have for Quinuas. Sentence starting with "PSD" should be moved under Materials and methods section, please see previous comments.

Soil variables used as predictors are mentioned in lines 125 f. (please see comment C13). Reasons, why PSD measurements were not possible for Quinuas, are explained in lines 132 f. (please see comment C8). The sentences in lines 315 ff. were adapted.

**C29:** L246: Why performance of Laipuna PTFs for pF0, pF 0.5 and pF 1.5 is lower that that of Quinuas? Please discuss how those could be improved.

We detected an error in Fig. 7 that was corrected:  R² values of the differential evolution pF 0 Laipuna models range from 0.64 to 0.78 and are similar to those achieved in Quinuas. No significant difference between models of both research areas can be detected based on the R² values and the scatter plots shown in Fig. 8 and 9.

It was discussed how to improve model performance in different sections of the paper (please see comment C36).

**C30:** L253-273: Please add title to that section, to highlight that you applied existing PTFs on the sites to compare the performance of the newly derived PTFs to those.

A new Section (3.3) was added under Results and discussion.

**C31:** L253: please add number of samples of the Laipuna dataset. Did you use the test set for the comparision?

Please see comment C37.

**C32:** L254: … PTFs from the literature were selected … Or add something similar.

"from the literature" was added to line 226.

**C33:** L254-256: please move it under Materials and methods.

The selection of existing PTFs to be applied on the Quinuas and Laipuna datasets was explained in a new Section (2.6) under Materials and methods.

**C34:** L256: it might be more precise to write that silt and sand content was converted to 2-50 _m and 50-2000 _m fractions by spline interpolation to calculate the USDA texture classes.

The sentences in lines 229 ff. were adapted.

**C35:** L263: add unit of RMSE.

Please see comment C37.

**C36:** L266-267: please mention other factors as well which could increase the performance of the PTFs.

The predictive performance of a model is affected by three factors: 1) The adjustment of the algorithm to the modelling problem by parameter tuning, which is discussed in Section 3.4. 2) The quality and size of the dataset forming the model input, which is discussed in lines 315 ff., 339 ff. and 460 f.. And 3) the chosen model algorithm, which was added to lines 342 f..

**C37:** L271: Table 2: - add number of samples – by pF values – used to test the newly derived and existing PTFs, did you use the test set of Laipuna dataset? - please use also here pF 0, 0.5, 1.5 and 2.5 instead of Theta 0, 0.5, 1.5 and 2.5., - add unit of the RMSE.

The same test sets were used to evaluate the performance of the newly developed PTFs and the existing PTFs. Using cross validation each sample was used for testing (please see Section 2.5). In Table 2 the numbers of response – predictor variable data pairs were added and "Theta" was changed to "pF". The RMSE values are shown without the unit (please see comment C16).

**C38:** L275-278: for easier comparison Figures 6 and 7 could be concatenated by using grouped boxplots.

The boxplots of Fig. 6 and Fig.7 were not concatenated as models built on datasets of two research areas with very different properties cannot be compared directly.

**C39:** L280-285: based on Figure 5 observed pF values of Quinuas site is greater then 0.30 cm3/cm3, for Laipuna those are greater than 0.20 cm3/cm3. Please check in calculations why you have observed pF values close to 0 cm3/cm3 on Figures 8 and 9. Or are those scaled observed and scaled predicted variables? It would be more informative to show the scatterplot for not scaled observed and predicted values. Please revise Figures 8 and 9.

Fig. 8 and 9 show scaled values, which was added to the figure captions. As predicted and observed values were scaled in the same way, the not scaled scatterplots would look like the scaled ones – except for the axis labels. Scaled values were used to be able to use the same axis (between 0 and 1) for each plot. It enhances comparability without losing information.

**C40:** L293-294: Sentence starting with "Difference": there is difference between GS and DE in case of the bag fraction as well. Is it possible to show which parameter – among number of trees, shrinkage, interaction depth, bag fraction – has the most dominant influence on the performance of BRT?

As mentioned in lines 430 ff. there is a difference in bag fraction, which cannot be explained (lines 446 f.). There might not even be an optimum for bag fraction (lines 444 f.).
We made an assumption about the parameter importance in lines 443 f.. A sensitivity analysis could be used to estimate the importance of each parameter. As the model performance depends on the

interaction of all parameter values (added to lines 442 f.), we recommend tuning the number of trees, shrinkage, interaction depth and bag fraction simultaneously.

**C41:** L318: … for the final differential evolution models derived for Laipuna site (Fig. 9) …
The sentence in lines 441 f. was adapted.

**C41:** L334, 339: In the caption of Figure 10 and 11: add number of tested parameter vectors for both method.
The number parameter vectors tested by grid search is mentioned in lines 170 f.. In accordance with the review of Mr. Wadoux the number of differential evolution iterations was added to lines 413 ff.. One iteration compared 100 parameter vectors (line 182).

**C42:** L343-344: please note that in most of the cases local PTFs perform better than PTFs trained on dataset originating from elsewhere with different soil forming factors. Please revise the sentence.
Please see lines 35 ff.. The comparison to other existing PTFs that were developed under conditions as similar as possible (lines 225 f.) is important to legitimate the developed BRT PTF and should be mentioned in the conclusions.

**C43:** L351-354: please consider to concatenate the last two sentence of the conclusions to better balance highlight both on the newly derived PTFs and results of comparing parameter tuning methods.
The conclusion was divided into two new paragraphs to highlight the two achievements. The last two sentences were concatenated (lines 484 f.)

**C44:** L354: please consider to provide availability of derived PTFs – which you recommend to use – for users.
The developed PTFs, as well as the underlying datasets, will be uploaded to the Open Science Framework (OSF) upon acceptance (added to lines 488 ff.).

**Technical comments**

**TC1:** L67: Please add the country after Páramo.
”Ecuador” was added after “Páramo” (line 82).

**TC2:** L151: The acronym of BRT is not included in the flowchart.
“BRT = boosted regression trees” was removed from line 195.

**TC3:** L343: … readily available …
„ready" was changed to „readily" (line 475).

**Referee comment 2**

The objective of the study is to develop pedotransfer function for water contents at four pressure heads (PF 0, PF 0.5, PF 1.5, PF 2.5) for two tropical mountain regions with high soil organic carbon content. Boosted regression tree technique was used to fit the models for both areas considering two tuning procedures to determine the regression tree-model parameters (n.tree, shrinkage, interation depth, bag fraction): grid search and differential evolution, the latter showing better results on the water retention estimates for the both areas. The work also compared the performance of the proposed PTFs with other PTFs from literature confirming the better performance of the proposed models. I congratulate the authors for the effort in collecting physico-chemical and hydrological data in such atypical soils and for using innovative techniques, such as the differential evolution, in order to get better results on the models fits. I also congratulate them for developing PTFs for organic soils which are not so common in the literature. The work in well written and structured and the subject is well posed. Some general and specific comments are summarized below:

**a)** Line 45: In organic Finnland soils?
Yes, in Finish peat soils. The sentence in line 47 was adapted.

**b)** It was not clear in the text why you have chosen the boosted regression tree models (Lines 72-73);
The reasons for using boosted regression tree models were added to lines 86 ff..

**c)** The sentence in line 97-98 should be reformulated ("It allows representing a research...to the accessible area"). The way it was written was unclear to me.
The description of the sampling site selection algorithm was extended (lines 115 ff.)

**d)** Line 105. Sometimes outliers should carry important information from the studied area. You should detail the reason of removing them.
The sentences in lines 134 ff. were adapted and extended to explain why it was decided to remove outliers.

**e)** The description of the boosted regression tree should be improved by describing clearer its fitting procedure (Lines 110-115).
The BRT fitting procedure was described more detailed (lines 152 ff.)

**f)** Line 144: What the word "respectively" is related to?
The sentence in lines 186 f. was adapted.

**g)** Line 182: After explaining the reasons for excluding the outliers it would be interesting to inform the range of their values for each soil property;
As only data pairs of response and predictor variables that were identified as multivariate outliers were removed (lines 135 f.), it is not useful to inform about the range for each soil property. The complete dataset, including data pairs identified as multivariate outliers, will be available upon acceptance (please see comment r).

**h)** Lines 182-190: What is right and left- skewed distribution? It is not clear.
The terms "right- and left- skew" were changed to the more common terms "negative- and positive-skew" (Section 3.1).  Negative skew: the mean is smaller than the median. Positive skew: the median is smaller than the mean.

**i)** Line 199: I suggest to correct this sentence: "..organic matter being characterized by a high water holding capacity" to *organic matter which is associated with soils with a high water holding capacity*
The sentence was corrected (lines 258 ff.).

**j)** Line 218: How the scaled water retention value was defined? This needs to be clarified;
Water retention values were scaled to the range [0, 1] using Eq. 1 (added to lines 138 ff.).

**k)** Line 230: Change Fig.11 and 12 to Fig 8 and 9;
Fig.11 and 12 were changed to Fig. 8 and 9 (lines 302 f.).

**l)** Line 240: Change Section 3.2 to Section 3.3;
Section 3.2 was changed to Section 3.3 (line 312).

**m)** Line 245: "PSD measurements were not included..in this area". This sentence should go to Section 3.1 when you call Fig.4 in the text.
The sentence in line 332 was deleted. Following the comments of the anonymous referee #1 , it was explained why PSD measurements were not possible for Quinuas (section 2.2, lines 124 ff.). The caption of Fig. 4 was adapted.

**n)** Line 234: Change Fig.9 a-d and 10 a-d to Fig.8 a-d to Fig.9 a-d;
Fig.9 and 10 were changed to Fig. 8 and 9 (lines 295 f.).

**o)** Lines 253-263: Did you apply the PTFs from the literature to the Laipura soils considering their range values applicability?
The test Laipura soils were included in the calibration of the proposed PTFs (BRT PTF – Table 2)? This need to be clarified.
Reliable PTFs, that were developed under conditions as similar as possible to the Laipuna dataset were selected (lines 225 f.). Selected PTFs are more general (line 16) as they were developed on larger datasets. The number of response – predictor variable data pairs was added to Table 2. However, the selected PTFs were not specifically developed for soils of Ecuadorian dry forest ecosystems and therefore response and predictors variable do not always match concerning their range.
The same test sets were used to evaluate the performance of the BRT PTFs and the existing PTFs. Using cross validation each sample was used for testing (please see Section 2.5).

**p)** Avoid vague sentences: ex: "these values are.."(Line 268), "in this case"(Line 320), "this might.." (line 320), "This might also result" (lines 325-326);
The sentences in lines 438-445 were adapted.

**q)** The code of the proposed models should be presented;
The developed PTFs, as well as the underlying datasets, will be uploaded to the Open Science Framework (OSF) upon acceptance (added to lines 489 ff.).

**r)** Is it possible to provide the study database to the readers?
Please see comment q.

**Manuscript with tracked changes**

[revised manuscript text omitted]